# The physiological basis for contrast opponency in motion computation in *Drosophila*

Giordano Ramos-Traslosheros [1,2] & Marion Silies [1✉]

In *Drosophila*, direction-selective neurons implement a mechanism of motion computation similar to cortical neurons, using contrast-opponent receptive fields with ON and OFF sub-fields. It is not clear how the presynaptic circuitry of direction-selective neurons in the OFF pathway supports this computation if all major inputs are OFF-rectified neurons. Here, we reveal the biological substrate for motion computation in the OFF pathway. Three inter-neurons, Tm2, Tm9 and CT1, provide information about ON stimuli to the OFF direction-selective neuron T5 across its receptive field, supporting a contrast-opponent receptive field organization. Consistent with its prominent role in motion detection, variability in Tm9 receptive field properties transfers to T5, and calcium decrements in Tm9 in response to ON stimuli persist across behavioral states, while spatial tuning is sharpened by active behavior. Together, our work shows how a key neuronal computation is implemented by its constituent neuronal circuit elements to ensure direction selectivity.

[1] Institute of Developmental Biology and Neurobiology, Johannes-Gutenberg University Mainz, Mainz, Germany. [2] International Max Planck Research School Neurosciences and Göttingen Graduate School for Neurosciences, Biophysics, and Molecular Biosciences (GGNB) at the University of Göttingen, Göttingen, Germany. ✉email: msilies@uni-mainz.de

The algorithms by which neuronal circuits detect motion using the visual input from photoreceptors is widely considered a paradigmatic neural computation. Similar computations to achieve direction selectivity, a hallmark of motion detection, appear to be implemented in visual circuitry as diverse as the *Drosophila* visual system and the vertebrate retina[1–3]. The neuronal circuits that extract direction-selective signals have been described in exquisite detail in *Drosophila*, perhaps the system in which we are closest to achieving a complete understanding of a neural computation from algorithm to circuitry and physiology[4–6]. However, despite recent work demonstrating that the *Drosophila* visual system implements a linear mechanism to compute motion[7], the implementation of this computation by the neuronal circuit elements has yet to be elucidated.

In *Drosophila*, direction selectivity emerges in the dendrites of T4 and T5 neurons, which respond to moving contrast increments (ON) and decrements (OFF), respectively[8,9]. These neurons are at least three synapses away from photoreceptors and receive inputs from distinct lamina and medulla interneurons of the ON and OFF pathways[10–13]. Two models, and their variants, have been used to describe direction-selective calcium signals in T4/T5[10,14,15–17]. These rely on nonlinear operations that either amplify signals moving in the detector's preferred direction[18] or suppress signals moving in the detector's null direction[19]. However, recent evidence based on electrophysiology and voltage imaging argues that direction selectivity can also emerge from linear summing of T4/T5 inputs[7], or nonlinearly through sublinear integration in a portion of the T4/T5 receptive fields resulting in null-direction suppression[20,21]. Specifically, sublinear voltage models are supported by experiments showing that direction-selective voltage signals in T4/T5 can be generated by summing synaptic input currents arising from individual bar responses of one contrast polarity[20,21]. In turn, a linear model predicted T5 voltage responses to moving sine wave gratings[7]. Furthermore, a dynamic nonlinearity in T4/T5 could adjust between these integration regimes[22]. In both models, the resulting voltage is then nonlinearly transformed into a calcium signal[17].

ON and OFF stimuli in the same location of the T5 receptive field produced voltage responses similar in amplitude, but opposite in sign[7]. This highlights the requirement for integration of ON and OFF stimuli across the T5 receptive field. However, the cellular substrates that support the linear summation of presynaptic inputs onto T5, across space and time, have not been described. Such a linear mechanism for direction selectivity parallels the one employed by simple cells in the vertebrate cortex of different species[23–26]. The receptive fields of simple cells in mammalian cortex have two subunits with spatially opponent inhibition, i.e., for the same contrast polarity one subunit is excitatory and the other is inhibitory[27,28]. Furthermore, cortical simple-cell receptive fields have a push-pull structure, meaning that at each receptive field location, inversion of the contrast polarity evokes a response of the opposite polarity, e.g., where ON stimuli provide excitation, OFF stimuli provide inhibition[27,29]. Similarly, T4/T5 neurons have putative excitatory (depolarizing) and inhibitory (hyperpolarizing) receptive field subunits that each show contrast-opponent responses, i.e., opposite responses to ON and OFF contrasts[7,17,20,21,30]. Thus direction-selective cells in the fly show contrast-opponent properties that resemble the push-pull structure of cortical simple-cell receptive fields[27]. Simple cell receptive fields are a consequence of the synaptic inputs received from the lateral geniculate nucleus (LGN) relay neurons as well as from other cortical cells, and T4/T5 receptive fields are set up by their presynaptic medulla cell inputs. Ultimately, to determine the physiological basis of linear summation in motion computation[7], the ON and OFF inputs to the direction-selective neurons need to be characterized.

Physiological studies of T4 and T5 response properties proposed that direction selectivity relies on inputs from three points in space (Fig. 1a, b)[15,16,31]. In both ON and OFF pathways, the medulla neurons that provide synaptic input to either the base or the tip of the dendrites have slower dynamics, whereas the central input is faster, consistent with the notion that direction selectivity requires temporal comparison of signals (Fig. 1a, b)[10,32,33]. Whereas T4 incorporates inputs from both, neurons that respond to ON and from neurons that respond to OFF signals, all major T5 inputs have been reported to be OFF rectified (Fig. 1b)[10,31–36]. Matching anatomy with the axis of motion, T5 receives its main visual inputs along the preferred motion direction axis from Tm9 on the trailing site, Tm1, Tm2, and Tm4 centrally, and CT1 on the leading site, with CT1 again mainly receiving Tm9 synaptic input (Fig. 1b)[13]. However, information about ON stimuli is required to achieve the contrast-opponent receptive field organization of T5, such that the excitatory subunit responds positively to OFF and negatively to ON, whereas the inhibitory subunit responds positively to ON and negatively to OFF[7] (Fig. 1c). Thus, it is not clear how T5 can receive the ON inputs required for its direction-selective receptive field.

Here, we investigate ON inputs, i.e., neurons that carry information about contrast increments, to T5 direction-selective cells in the OFF pathway to determine the cellular basis for the implementation of direction selectivity in T5 neurons. In vivo two-photon microscopy, dual-color calcium imaging, and characterization of all major OFF pathway interneurons reveal that Tm9, Tm2, and CT1 provide ON information to direction-selective T5 cells in the OFF pathway. Thus, these three neurons provide the basis for the T5 contrast-opponent receptive field subunits together with other OFF pathway neurons. Focusing on T5's major input, we show that receptive field properties of Tm9 are variable in size, and this variability correlates with T5 properties, thus demonstrating that Tm9 shapes downstream computations. Finally, we test whether these properties are stable across behavioral states, and whether motion computation relies on the same input properties. Mimicking an active behavioral state by activating octopamine receptors shows that both Tm9 variability and ON receptive field are maintained across states, whereas the Tm9 spatial receptive field is sharpened. Together, these findings reveal the cellular substrate of motion computation in the *Drosophila* OFF pathway and highlight the concrete neuronal properties that are responsible for direction selectivity.

## Results

**The presence of both ON and OFF inputs improves responses to motion**. A recent study showed that direction selectivity in the T4 ON pathway can be modeled by combining three inputs providing slow OFF inhibition, fast ON excitation, and slow ON inhibition (Fig. 1a)[37]. In the OFF pathway, an analogous model would require an ON inhibitory input on the trailing site of the receptive field corresponding to the dendrite base, where T5 receives most synaptic inputs from Tm9. To test whether direction selectivity in T5 would indeed benefit from yet uncharacterized ON inputs, we modeled responses to sine gratings moving in different directions by simulating the T5 dendrite as a passive cable that integrates inputs from the main presynaptic neuron types, Tm9, Tm1, Tm2, Tm4, and CT1. At each of the three input sites on the hexagonal ommatidia grid, a neuron's receptive field was represented with a linear-nonlinear (LN) model consisting of linear spatial and temporal filters[32], a synaptic function that transformed the filter response into a conductance change, an output nonlinearity, and an offset applied to prevent negative conductances (Fig. 1d). The output of the LN model was fed into the corresponding site in the passive

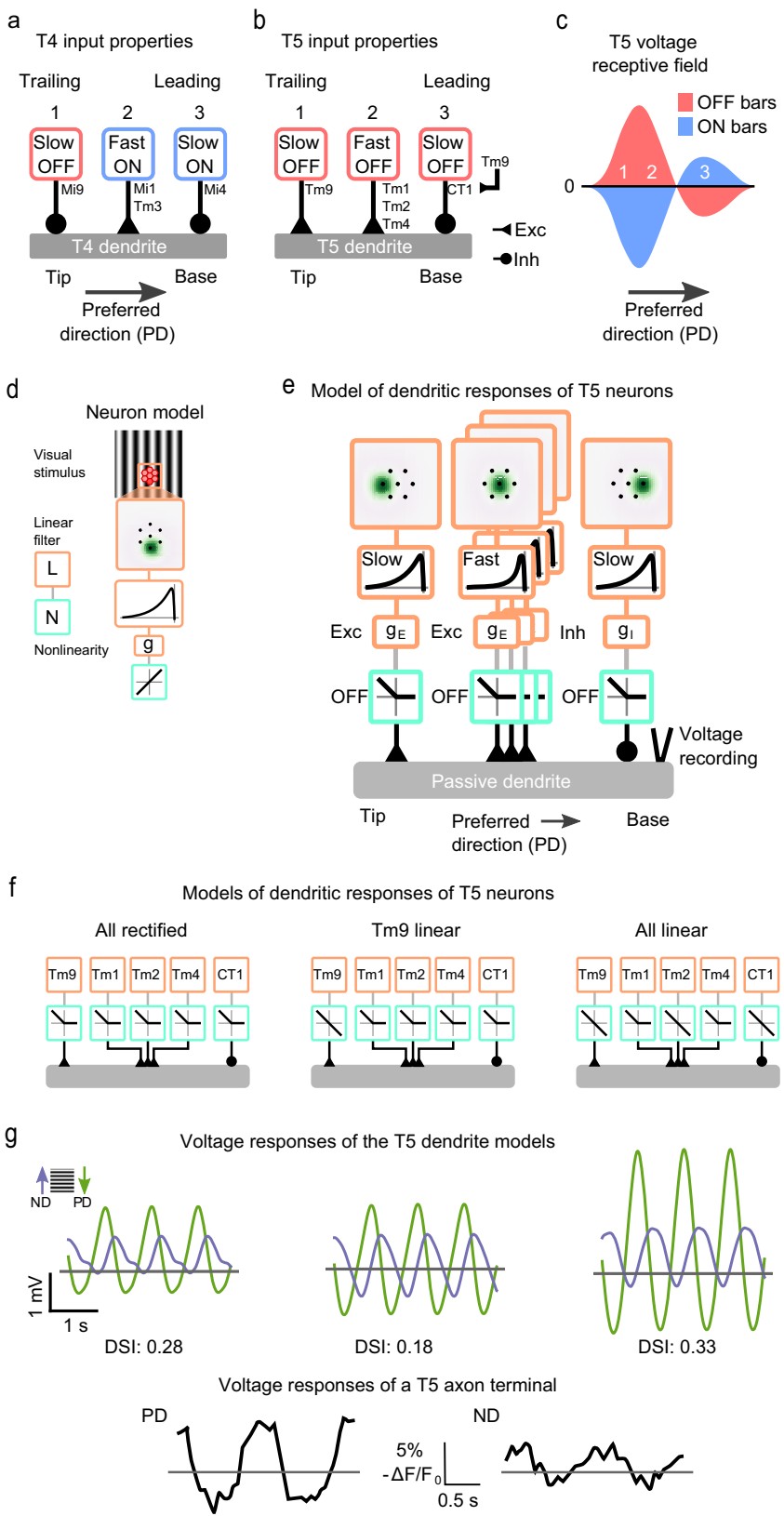

dendrite for each neuron type, weighted proportionally to the synaptic connectivity of each neuron type[13], the conductance inputs were transformed to input currents that were linearly added and integrated to obtain the membrane voltage[20,21,37]. Finally, the output voltage response was recorded from the base of the T5 model dendrite (Fig. 1e).

We considered several models differing in the response properties of the T5 inputs and compared the models on the basis of their direction selectivity computed across 16 motion directions (Supplementary Fig. 1), and show their response traces in the preferred and null directions (Fig. 1f, g). A model that received only OFF-rectified inputs produced direction-selective

**Fig. 1 Circuit organization and models of local motion detection in *Drosophila*. a** Schematic showing the dendrites of T4 neurons in the ON motion pathway. Inputs are organized from tip to base, corresponding to the trailing and leading site of the neuron's receptive field with respect to its preferred direction (PD): Mi9 (trailing, 1), Mi1 and Tm3 (central, 2), and Mi4 (leading, 3). Boxes show visual response properties. **b** Schematic showing the dendrites of T5 neurons in the OFF motion pathway. Inputs are organized from tip to base, matched to the visual sites in **a** as Tm9, Tm1/Tm2/Tm4, and CT1. **c** Schematic showing the receptive field organization of T5 neurons following Wienecke et al.[7]. **d** A linear-nonlinear model simulates the responses of T5 input neurons (linear, orange; nonlinear, light green). A moving sine wave grating is processed by a hexagonal array of linear spatial (dark green spot) and temporal filters (black line), a conductance function g and an output nonlinearity. The array matches the arrangement of units in the fly eye (red). **e** A full model of a T5 dendrite. The input organization and properties follow those shown in **b** using the model components described in **d**. All inputs are integrated by a passive dendrite, and the voltage response is recorded at its base. **f** Schematic showing T5 models: a model with all inputs OFF rectified, a model with a linear OFF input on the Tm9 site, and a model with all input sites receiving one linear OFF input. **g** Voltage responses in the preferred (green) and null (purple) directions (ND) to a sinewave grating from the models in **f** (top), and from Wienecke at al.[7] (bottom). Model responses are to a grating with a spatial frequency of 1/24 cycles/deg and temporal frequency of 1 Hz. The direction-selectivity index (DSI) is shown below the traces, calculated from the amplitude of the Fourier component at 1 Hz of the voltage responses for each of the 16 directions of motion simulated (Supplementary Fig. 1c).

responses reproducing a mechanism consistent with the one described in Gruntman et al.[21], which requires OFF excitation over the trailing and central sites (here Tm9, Tm1, Tm2, and Tm4), and OFF inhibition in a smaller region on the leading site (here CT1). However, this model failed to reproduce the sinusoidal responses of T5 neurons consisting of depolarizations to OFF and hyperpolarizations to ON grating components (Fig. 1f, g)[7]. Because of Tm9's position on the trailing site of the T5 dendrite (Fig. 1b), we next provided T5 with a linear input by using a linear output function for the excitatory Tm9 input. This way, we kept the known OFF response of Tm9 while still adding an hyperpolarizing ON component to the trailing site (Fig. 1f). The corresponding model responses had both ON hyperpolarizations and OFF depolarizations similar to biological T5 responses, but this model's direction selectivity was lower than the previous one (Fig. 1f, g), likely because it lacked ON inputs in the central and leading sites implementing the contrast-opponent properties of a T5 receptive field. A model with an additional linear output function for Tm2 in the central site produced qualitatively similar voltage responses, but improved direction selectivity for calcium responses (Supplementary Fig. 1a, b).

To address the potential benefit of ON inputs across the receptive field, we set linear output functions for Tm9 in the trailing site, Tm2 in the central site and CT1 in the leading site (Fig. 1f). This model produced responses that were both direction selective and biologically plausible in that they had a sinusoidal shape with both hyperpolarizations to ON and depolarizations to OFF (Fig. 1f, g). The previous conclusions hold across a range of parameters and stimulus conditions (Supplementary Fig. 1). Thus, even a situation in which just one of the inputs is linear is already sufficient to produce T5 responses with ON and OFF components, but only one linear input at each dendritic site produces qualitatively similar responses with higher direction selectivity. Altogether, the model used here hints that both nonlinear[21] and linear models[7] of direction selectivity in T5 can benefit from ON information. This suggests that additional ON information from T5 presynaptic neurons might be present.

**Tm9 provides ON information to T5.** Tm9 has been shown to have a central role in motion computation. It provides most synaptic input to T5, constitutes the sole input on the trailing site of the T5 receptive field, provides major input to CT1 on the leading site, and it leads to the strongest phenotypes when silenced alone or in combination with other Tm neurons[13,14,34]. However, Tm9 neurons have been previously regarded as OFF rectified[32,34]. To understand the origin of the ON input motivated by models of motion computation[7] (Fig. 1), we first characterized Tm9 receptive field properties to both ON and OFF stimuli using in vivo two-photon calcium imaging. We expressed the genetically encoded calcium indicator GCaMP6f in Tm9 neurons, and

recorded calcium signals in response to visual stimuli from Tm9 axon terminals in the first layer of the lobula (Fig. 2a).

To probe Tm9 spatial receptive fields we used ON bars (bright bars on a dark background) and OFF bars (dark bars on a bright background). Individual five-degree bars were shown as 0.5 Hz flicker. Tm9 neurons responded positively to OFF bars within a constrained region on the screen covering ~6.9° (Fig. 2b, c and Supplementary Fig. 2a, b). Interestingly, Tm9 neurons responded with a decrease in calcium to ON bars (Fig. 2b). Individual Tm9 axon terminals responded negatively to ON bars covering a wider spatial range and sometimes spanning the whole stimulation area (Fig. 2b and Supplementary Fig. 2c, d). Quantification of the spatial receptive field showed that Tm9 ON responses were variable, ranging from 8.3° to 71.5°. Negative Tm9 responses to ON bars (ON receptive field) were generally wider than positive Tm9 responses to OFF bars (OFF receptive field), both for horizontally and vertically oriented bars (Fig. 2b, c). In summary, Tm9 neurons exhibit both an ON and an OFF receptive field component that differ in their spatial extent.

This larger ON receptive field size could account for a discrepancy in previous studies that reported narrow receptive fields observed with OFF bars[34], and wide receptive fields measured with white noise[14], which could be explained by the differential activation of distinct ON and OFF receptive fields. White noise, containing both bright and dark bars, may activate the wide ON component of the receptive fields. In another experimental set, we recorded Tm9 receptive fields for the same neurons using ON and OFF bars as well as ternary white noise (white, gray, or black stripes). To compare spatial receptive field properties across stimuli, we represented the two-dimensional receptive field as a sign-consistent Cartesian product of the one-dimensional receptive fields obtained from horizontal and vertical stimulus presentations (Fig. 2d). The receptive fields measured with ternary white noise were larger than the OFF but smaller than the ON receptive fields, seen in examples from six different flies, as well as in the population average (Fig. 2e, f). Thus, stimuli that contain ON components elicit wider receptive fields than pure OFF stimuli, and there is some interaction between the two.

To investigate whether the measured receptive field properties were specific to Tm9, we first simultaneously measured the activity of Tm9 and another Tm neuron, Tm4. To record the activity of two neurons responding to the same point in visual space, we used different binary expression systems to express GCaMP6f specifically in Tm9 neurons and the red-shifted calcium indicator jRGECO1a in Tm4 (Fig. 3a). We further picked Tm4 because of its spatial separation from Tm9 terminals[12]. Tm9 and Tm4 both responded to full field OFF flashes with sustained and transient kinetics, respectively. However, Tm9 responded more strongly to the gray-OFF than to the ON-gray transition, consistent with Tm9 receiving input

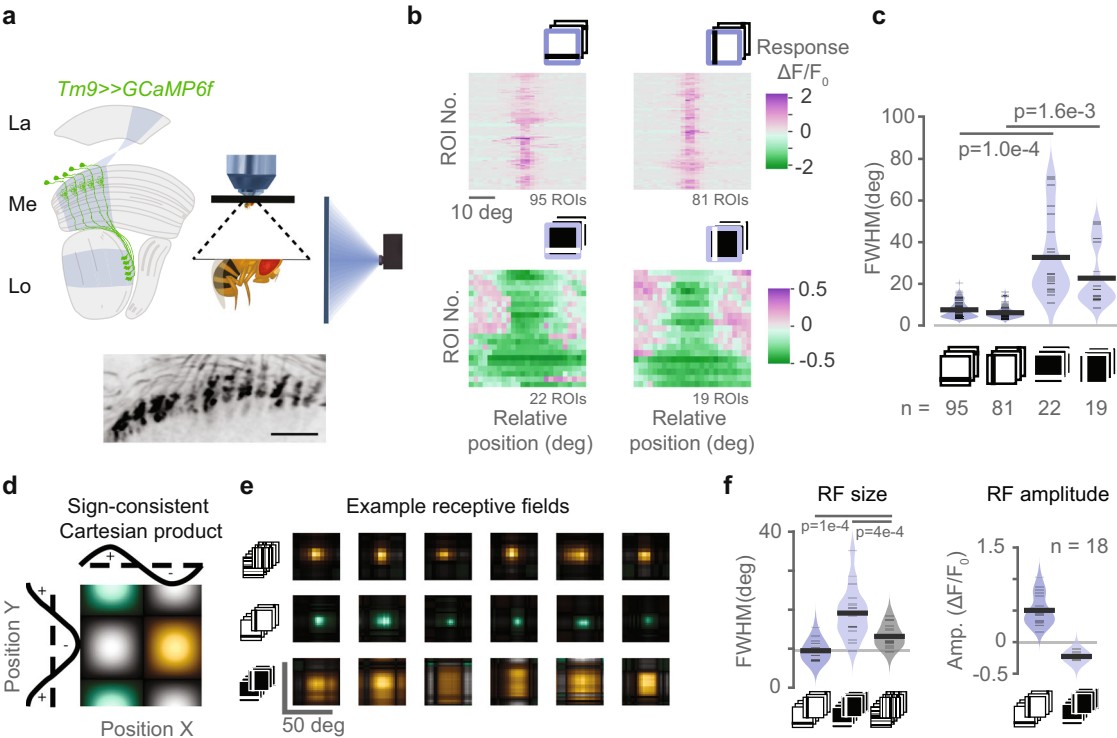

**Fig. 2 Tm9 has wide ON and narrow OFF receptive fields. a** Schematic showing the experimental setup used for in vivo two-photon calcium imaging of Tm9 neurons expressing GCaMP6f under visual stimulation. Visual information (blue) passes from the lamina (La) through the medulla (Me) housing Tm9 neurons into the lobula (Lo), where we recorded from Tm9 axon terminals (example image from a recording shown at the bottom, 1 from 14 flies). Scale bar = 10 μm. **b** Receptive fields obtained from Tm9 responses to ON (blue) and OFF bars (light blue) in horizontal and vertical orientations. Receptive field centers are aligned for illustrative purposes. For each stimulus, only neurons with responses fitted by a single Gaussian with $r^2 > 0.2$ and response quality index above 0.5 were included. **c** Full width at half maximum (FWHM) of a single Gaussian fitted to the receptive fields in **b**. $n$ = 95, 81, 22, 19 ROIs (regions of interest), from 12, 10, 6, 5 flies. **d** Two-dimensional visualization of spatial receptive fields by color-coding the Cartesian product of two orthogonal one-dimensional receptive fields (from calcium signal or filter amplitude) according to the sign and response strength: positive in both orientations in green, negative in both orientations in yellow, and opposing signs in gray. Color luminance linearly encodes the magnitude of the Cartesian product. **e** Receptive fields obtained with different stimuli for the same neurons: ternary white noise (top), OFF bars (middle), and ON bars (bottom). Panel shows receptive fields from six example neurons from six flies. **f** Amplitude and FWHM obtained from fitting a single Gaussian to each neuron's receptive field in the population ($n$ = 18 ROIs, from six flies). **c**, **f** p-values come from two-tailed permutation tests. Source data are provided as a Source Data file.

from the luminance-sensitive L3 neuron[38]. Tm4 did not respond to ON flashes, whereas Tm9 showed weak decreases in calcium signal in response to full field ON flashes (Fig. 3b). This difference became more prominent when extracting receptive fields from local bar stimuli. No responses to bright bars were detectable in Tm4 neurons, whereas Tm9 responded negatively to narrow bright bars, and displayed a wide ON receptive field (Fig. 3c–e and Supplementary Fig. 3). This was visible in individual ROI responses (Fig. 3c) as well as in the population average (Fig. 3d). Spatial OFF receptive fields were narrow for both Tm9 and Tm4, consistent with previous reports[34] (Fig. 3c–e). Since Tm9 responses in the receptive field center reverse in sign upon reversing the stimulus polarity, one can think of them as having a receptive field structure resembling that of the LGN inputs to simple cells in mammalian V1. The existence of a negative ON receptive field suggests that Tm9 is a suitable candidate to provide the neural substrate for the ON T5 input implementing a linear mechanism of direction selectivity on the trailing site of the T5 receptive field.

**Tm2 and CT1 provide central and trailing ON inputs to T5.** In addition to the Tm9 ON receptive field, ON inputs at the central and leading sites can explain the proposed linear mechanism of direction selectivity[7] (Fig. 1). On the central site, T5 gets most of the inputs from Tm1, Tm2, and Tm4[13]. To test whether any of

those could provide a central ON input to T5, we characterized their ON receptive field properties using narrow bars. For a comprehensive analysis of all Tm neurons, we expressed GCaMP6f in Tm1, Tm2, Tm4, or Tm9, and recorded calcium signals in response to ON and OFF bars from the axon terminals of each neuron type (Fig. 4a and Supplementary Fig. 4). All Tm neurons responded positively to OFF bars within a constrained region on the screen, consistent with data shown in [34] (Fig. 4b–e, j, k). Tm4 again did not show an ON receptive field, this time imaged with GCaMP6f instead of jRGECO1a, showing that this is not due to the indicator used. Tm1 also did not respond to ON bars. Interestingly, both Tm9 and Tm2 neurons responded negatively to ON bars with a mean full width at half maximum (FWHM) of 13 deg (Tm2), and 17 deg (Tm9) (Fig. 4g, i, l, m). Both ON and OFF bar responses were similar for horizontally and vertically oriented bars. Therefore, Tm2 has both an ON and an OFF receptive field component that differ in their spatial extent. Altogether, our results show that Tm2 can provide the ON input to the central site of the T5 receptive field required by recent data demonstrating linear summation[7], and that additionally improves direction-selectivity in nonlinear models (Fig. 1).

The T5 leading site gets most of its inputs from CT1 neurons, which in turn receive most of their inputs from Tm9[13]. Thus, the ON input from Tm9 can in principle be conveyed to the leading site through CT1. Furthermore, CT1 is thought to provide

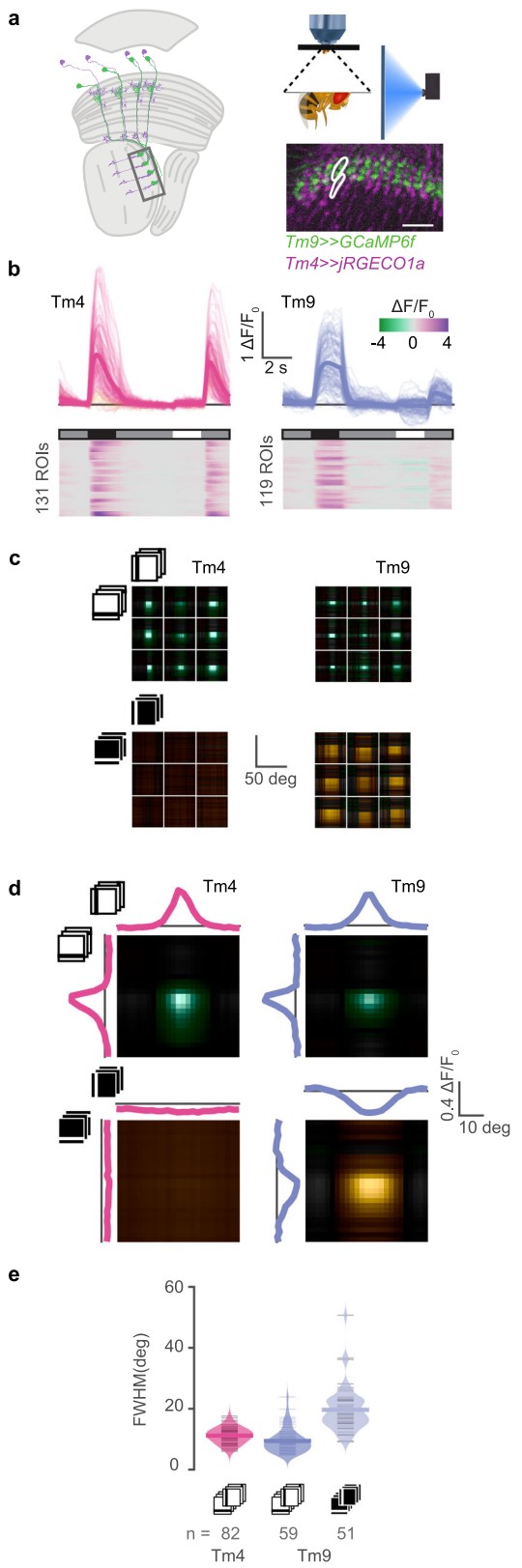

**Fig. 3 Dual-color imaging reveals specificity of the Tm9 ON receptive field. a** Experimental setup used for in vivo two-photon calcium imaging under visual stimulation of Tm9 neurons expressing GCaMP6f (green), and Tm4 neurons expressing jRGECO1a (magenta). Scale bar = 10 μm. **b** Responses to full-field ON and OFF flashes from an intermediate gray background. Both Tm9 (blue) and Tm4 (pink) respond preferentially to OFF transitions, but only Tm9 responds (negatively) to ON flashes. Thick lines are means across ROIs, thin lines are single ROIs. **c** Receptive fields of nine example neurons from each neuron type. Only Tm9 has ON receptive fields. **d** Population average of receptive fields simultaneously recorded and aligned to the center for visualization by maximizing their cross-correlation. Both Tm9 and Tm4 have narrow OFF receptive fields but only Tm9 has a wide ON receptive field. **e** Full width at half maximum (FWHM) obtained from fitting a single Gaussian to each neuron's receptive field in the population. Tm9 ON (light blue, $n = 51$ ROIs), Tm9 OFF (blue, $n = 59$ ROIs), and Tm4 OFF (pink, $n = 82$ ROIs). ROIs come from 11 flies. Tm4 did not have an ON receptive field. Values are averages over the two orientations. Only neurons with OFF receptive field fits with $r^2 > 0.8$ for both orientations were included. Source data are provided as a Source Data file.

compartmentalization shown in a previous study[36]. Moreover, CT1 responded positively to ON bars in the locations surrounding the OFF receptive fields, and decreased responses in the location where it was excited by OFF bars (Fig. 5c, d). Because CT1 responds positively to OFF stimuli and negatively to ON, it can not only provide T5 with the asymmetric OFF inhibition supported by nonlinear models[21], but also the complementary, disinhibitory ON input supported by linear, contrast-opponent models (Fig. 1b)[7]. Therefore, T5 can access ON information throughout its receptive field.

**Tm9 variability correlates with T5 tuning properties**. We next asked whether Tm9 properties shape downstream T5 computations. To do this, we harnessed the variability observed in Tm9 responses (Fig. 2c), and asked whether this is passed on to downstream computations. We focused on this neuron again, because blocking Tm9 had the strongest effect on direction-selective responses downstream[14,34], and because of Tm9 being T5's most numerous input. We simultaneously recorded responses from Tm9 axon terminals and downstream T5 dendrites and asked whether Tm9 variability is reflected in the properties of downstream T5 neurons. To do so, we expressed jRGECO1a in Tm9 and GCaMP6f in T5 and recorded signals in the lobula, where Tm9 axon terminals synapse onto T5 dendrites (Fig. 6a). Since T5 neurons exist in four subtypes, each selective for one of the four cardinal directions of motion, each Tm9 axon terminal contacts the dendrites of four different T5 neurons. By virtue of T5 orientation selectivity[8], horizontal stimuli predominantly stimulate the two T5 subtypes selective for upward and downward motions, and vertical stimuli predominantly excite front-to-back and back-to-front selective T5 subtypes.

We first recorded receptive fields using ON and OFF bars. Both Tm9 and T5 responded positively to OFF bars, whereas only Tm9 showed prominent negative responses to ON bars, demonstrating that we can spectrally separate Tm9 and T5 signals (Fig. 6b–e). We extracted the OFF receptive field properties from single Gaussian fits to the tuning curves. A tight correlation for the receptive field position confirmed that Tm9 and T5 neurons, that were recorded together were responding to the same point in visual space (Supplementary Fig. 5a, b). Interestingly, the widths of T5 and Tm9 OFF receptive field were positively correlated, indicating that a larger Tm9 receptive field

GABAergic inhibition to T5[11]. To test what spatial contrast information CT1 provides to T5, we measured responses to ON and OFF bars from axon terminals in the lobula layer 1 of CT1 neurons expressing GCaMP6f (Fig. 5a). CT1 showed positive, spatially confined responses to OFF bars with a FWHM of approximately 5° (Fig. 5b), consistent with its strong

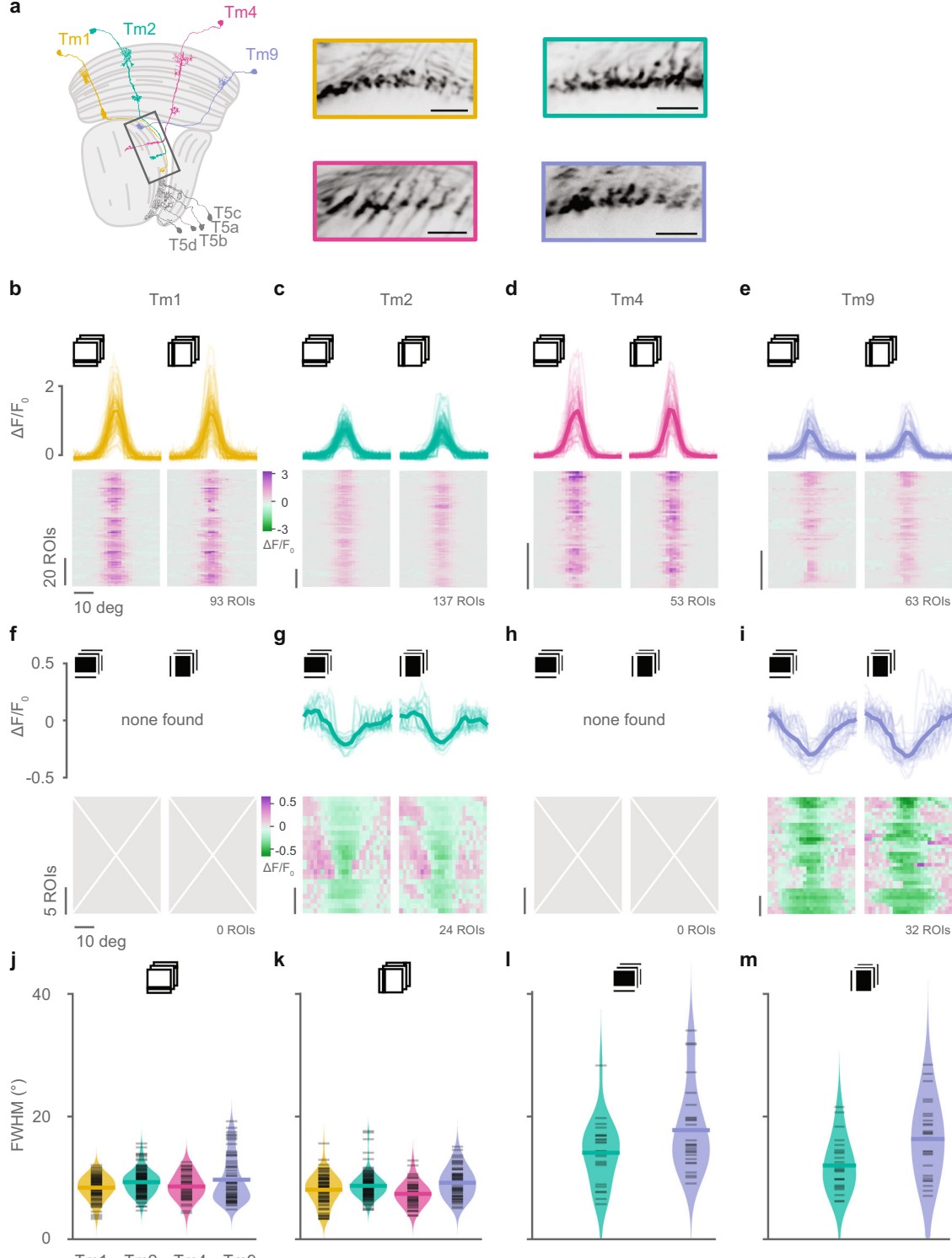

**Fig. 4 Tm2 and Tm9 neurons have ON receptive fields. a** Schematic of the optic lobe showing the anatomy of the main medulla input neurons to T5, Tm1, Tm2, Tm4, and Tm9, and example images from recordings of their axon terminals. Scale bar = 10 μm. **b–e** OFF receptive fields obtained from in vivo two-photon calcium imaging of responses to horizontal and vertical dark bars from Tm1 (**b** yellow), Tm2 (**c** green), Tm4 (**d** pink), and Tm9 (**e** blue) neurons expressing GCaMP6f. Only neurons with responses in both orientations fitted by a single Gaussian with $r^2 > 0.2$ and response quality index above 0.5 were included. Number of ROIs (neurons) is shown below the plots on the right. Thick lines are means across ROIs, thin lines are single ROIs. **f–i** ON receptive fields obtained from responses to horizontal and vertical bright bars from Tm1 (**f**), Tm2 (**g**), Tm4 (**h**), or Tm9 (**i**) neurons in **b–e**. Only Tm2 and Tm9 neurons had ON receptive fields. **j–m** Full width at half maximum (FWHM) from receptive fields obtained from horizontal OFF bars (**j**), vertical OFF bars (**k**), horizontal ON bars (**l**), and vertical ON bars (**m**). Data points correspond to neurons in **b–i** matching the stimulus condition. Sample sizes are Tm1 ($n = $ 93 ROIs), Tm2 ($n = $ 137 ROIs), Tm4 ($n = $ 53 ROIs), and Tm9 ($n = $ 63). ROIs come from 11 (Tm1), 15 (Tm2), 8 (Tm4), and 11 (Tm9) flies for all panels. Source data are provided as a Source Data file.

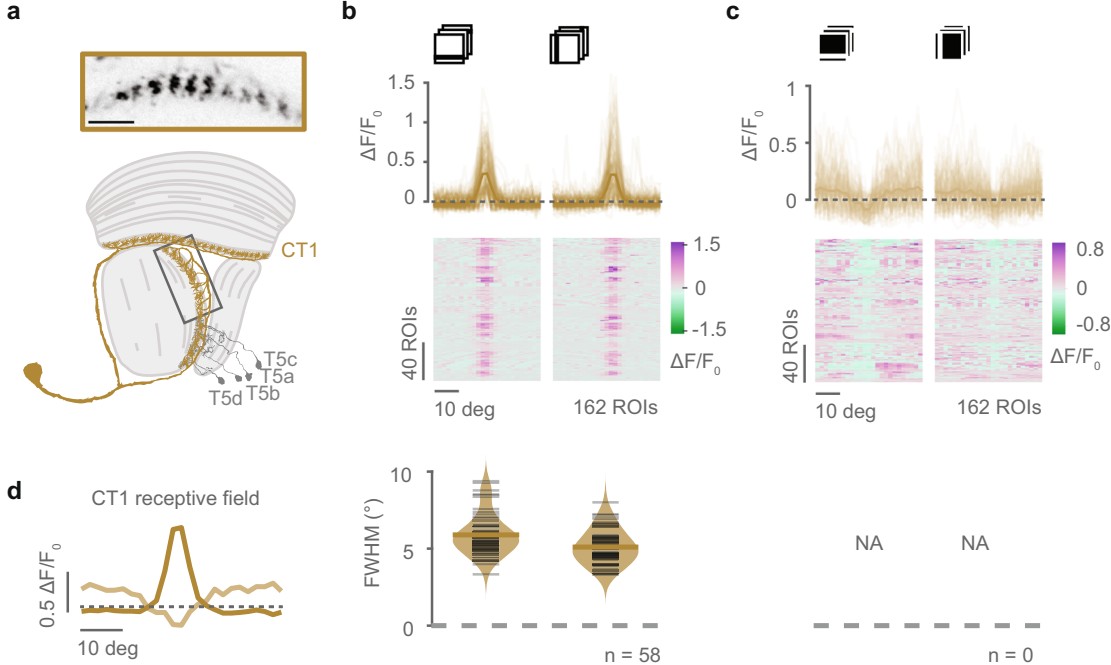

**Fig. 5 CT1 neuron responses contain ON and OFF information. a** Example average intensity projection from recordings of axon terminals from a single CT1 neuron (top), and a schematic of the optic lobe showing the anatomy of the CT1 neuron (bottom). Scale bar = 10 μm. **b** Top: OFF receptive fields obtained from in vivo two-photon calcium imaging of responses to horizontal and vertical dark bars from CT1 neurons expressing GCaMP6f. Shown are all recorded ROIs (n = 162, thin brown lines, from 15 flies) and mean across ROIs (thick brown lines). Bottom: Full width at half maximum (FWHM) from receptive fields obtained from horizontal and vertical OFF bars from ROIs in top panel fitted by a Gaussian. Only neurons (ROIs) with response quality index above 0.5 and $r^2 > 0.2$ are shown, similarly to Fig. 4j–m. n = 58 ROIs from 15 flies. **c** As in **b** but for ON receptive fields. Top: CT1 responses to horizontal and vertical ON bars in light brown. Bottom: The CT1 ON receptive fields could not be well-fitted by a Gaussian. **d** Average receptive fields from vertical OFF and ON bars from **b**, **c** overlaid to visualize the spatial organization of CT1 responses to ON and OFF bars. n = 162 ROIs from 15 flies. Source data are provided as a Source Data file.

generates a larger T5 receptive field (Fig. 6f). This suggests that the variability observed in receptive fields is passed on to downstream computation and confirms tight coupling between Tm9 and T5 properties. To test whether this functional correlation extends to moving stimuli that include both ON and OFF components, we next measured Tm9 and T5 responses to moving sine wave gratings. We measured the spatiotemporal frequency tuning of these neurons using gratings of different spatial and temporal frequencies. Gratings were moving back-to-front to selectively stimulate one subtype of T5 selective for that motion direction. We quantified the spatiotemporal frequency tuning of a neuron using the response amplitude of the Fourier component matching the temporal frequency of the grating. Average tuning properties across the population of all recorded Tm9 and T5 cells were highly similar both spatially and temporally (Fig. 6g). Comparing the tuning between individual Tm9 and T5 neuron pairs showed a strong pairwise correlation with a mean correlation coefficient of 0.8 (Fig. 6h). Overall, Tm9 and T5 response properties are correlated for static stimuli, as well as for moving stimuli with both ON and OFF components. This functional link further supports Tm9's important role in the computation of direction selectivity by T5, and the biological relevance of variability mediated through Tm9.

**Octopamine signaling sharpens the ON receptive field of Tm9.** The tuning of visual behaviors and of direction-selective cells depends on the behavioral state of the animal[32,39–42]. Tm9 is required for direction-selective responses across a wide range of speeds[14,34]. This suggests that Tm9 properties should be

maintained across behavioral states, such as flying or walking, where the scene moves at faster speeds relative to the fly. Furthermore, to accurately compute local motion cues, it appears disadvantageous to sample across larger regions of visual space. We hypothesize that Tm9 neurons should display both negative ON and positive OFF responses across behavioral states, but these responses should be sharpened when the animal is walking or flying. To test this hypothesis, we recorded from Tm9 upon application of chlordimeform (CDM), an octopamine agonist that mimics active behavioral states and modulates the gain and selectivity of direction-selective cells[32,39–43]. Tm9 responded negatively to ON and positively to OFF bar stimulus polarities with or without CDM (Fig. 7a, b). CDM appeared to decrease OFF center responses and strengthen the surround (Fig. 7a). To quantify the center-surround interaction, we fitted a difference-of-Gaussians to the receptive fields. Indeed, CDM application significantly decreased the OFF receptive field center amplitude (Fig. 7c and Supplementary Fig. 6a). Both center and surround amplitudes of the ON receptive fields were reduced upon CDM application (Fig. 7c), while absolute minimum and maximum responses did not change (Supplementary Fig. 6a). Furthermore, the OFF receptive field surround and the ON receptive field center became significantly sharper (lower FWHM) in the presence of CDM (Fig. 7d), resulting in sharpened net receptive fields (Supplementary Fig. 6b), whereas the variability of Tm9 responses was not affected by CDM. Comparing center and surround in individual neurons showed that CDM did not change correlations of center-surround amplitudes for OFF bars and reduced them for ON bars (Fig. 7e).

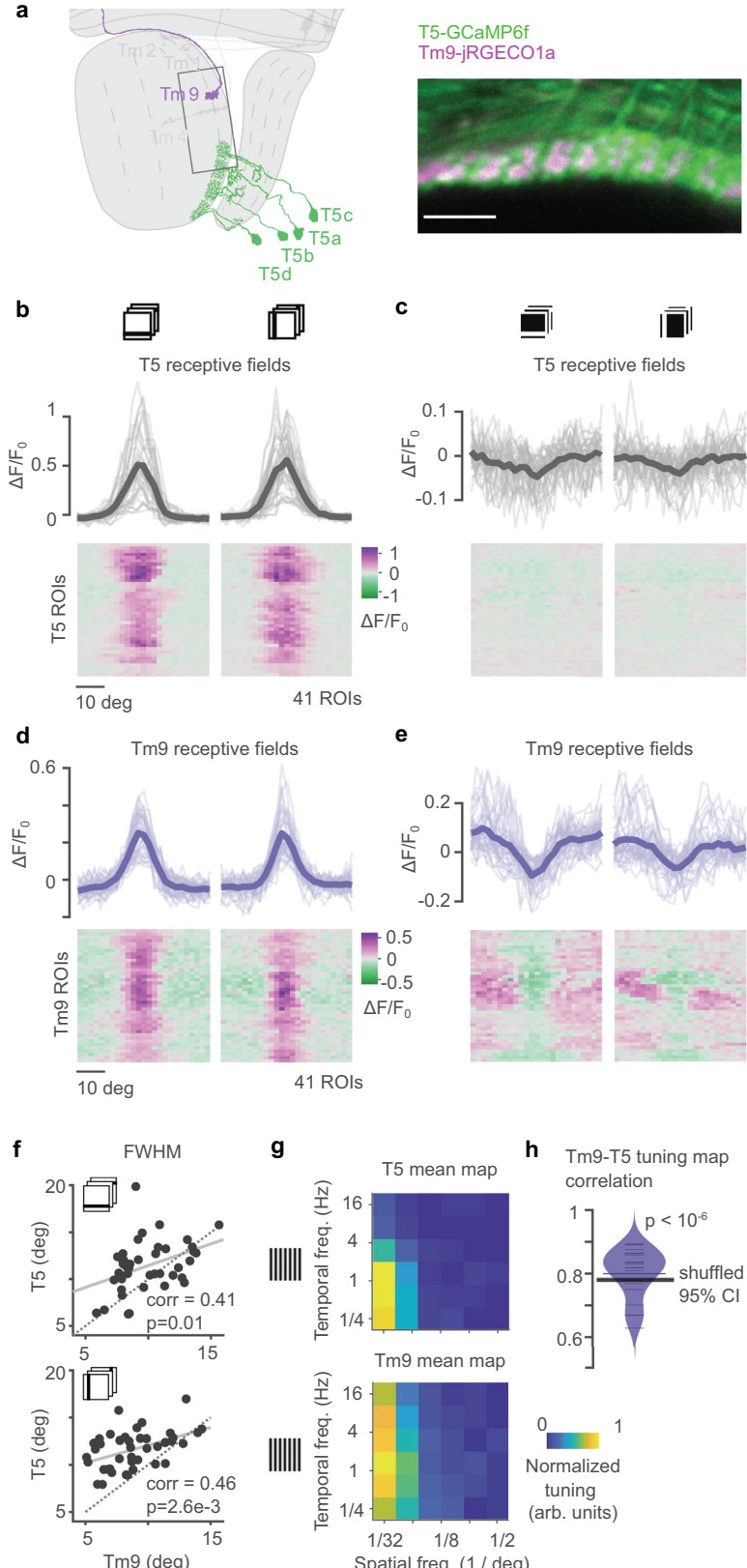

This center-surround organization in the presence of CDM suggests that responses to global OFF stimuli might be suppressed. When we measured Tm9 responses to full-field flashes, we indeed observed that Tm9 responses decreased to OFF flashes (Fig. 7f, g). Conversely, Tm9 negative responses to full-field ON stimuli were unchanged, again consistent with the net center-surround

amplitude of ON receptive fields, that did not change in the presence of CDM (Fig. 7f, g and Supplementary Fig. 6a). Taken together, our data show that octopamine signaling, mimicking an active behavioral state, sharpens the spatial receptive field of Tm9 neurons, while maintaining both negative ON and positive OFF components for motion estimation[7].

**Fig. 6 Tm9 variability correlates with T5 functional properties. a** Schematic showing the proximal fly optic lobe. Inset shows an example image from a recording of T5 neurons expressing GCaMP6f (green) and Tm9 neurons expressing jRGECO1a (magenta). Scale bar = 10 μm. **b, c** OFF (**b**) and ON (**c**) receptive fields obtained from in vivo two-photon calcium imaging from responses to horizontal and vertical dark bars from T5 neurons expressing GCaMP6f. Same neurons for ON and OFF ($n = 41$ ROIs from ten flies). T5 did not respond to ON bars. Thick, gray lines are means across ROIs, thin, gray lines are single ROIs. Only neurons with responses to OFF bars in both orientations fitted by a single Gaussian with $r^2 > 0.25$ and response quality index above 0.3 were included. **d, e** OFF (**d**) and ON (**e**) receptive fields of Tm9 neurons (blue lines) expressing jRGECO1a imaged simultaneously and overlapping with the T5 dendrites in **b, c** ($n = 41$). **f** Full width at half maximum (FWHM) for OFF receptive fields of Tm9 vs. T5 signals for horizontal (top) and vertical bars (bottom). Tm9 and T5 signals originate from the same region of interest. Pearson's correlations (corr), and associated two-tailed p-value ($p$), $n = 41$ ROIs (regions of interest). Solid line is the least-squares fit. T5 signals had greater FWHM than Tm9 signals, as seen by the points lying above the identity line (dotted). **g** Average spatiotemporal tuning maps for T5 (top) and Tm9 (bottom). Maps were obtained from the Fourier amplitude component matching the temporal frequency of the sinewave gratings of different spatial and temporal frequencies moving back-to-front. Maps were scaled by their maximum response before averaging. Both Tm9 and T5 were selective for similar spatial and temporal frequencies. $n = 21$ ROIs from ten flies. **h** Distribution of pairwise correlations between the spatiotemporal maps of T5 and Tm9 signals. $n = 21$ ROIs from ten flies. Mean correlation is about 0.8. To test whether pairwise correlations were sensitive to the particular pairs, we performed 1e6 random shuffles of the pairings of Tm9-T5 signals and quantified the 95% confidence interval of the mean correlation per shuffle. No random shuffle showed a larger mean correlation than the original mean correlation ($p < 10^{-6}$). Thus, the mean correlation is above the upper 95% confidence bound and confirms that T5 neurons are more correlated with their Tm9 input neurons than with other Tm9 neurons. Source data are provided as a Source Data file.

## Discussion

In this study, we sketched a minimal model that exposed the benefit of ON information to T5 and identified the ON inputs that provide the cellular basis to implement a mechanism for direction selectivity in the fly OFF motion pathway. Specifically, three OFF pathway interneurons, Tm9, Tm2, and CT1 have negative ON receptive field components, providing ON information to different spatial locations of the T5 spatial receptive field (Fig. 7h, i). Tm9 exhibits variability in its physiological receptive field properties, which are tightly coupled with downstream T5 properties. The ON properties of Tm9 are maintained in the presence of an octopamine receptor agonist, whereas spatial tuning of Tm9 is sharpened, matching constraints imposed by an active behavioral state.

**The neural basis for mechanisms of direction selectivity in the OFF pathway**. Our study provides critical insights into the physiological implementation of direction selectivity, and thus synthesizes different observations into a single mechanism: direction selectivity emerges from the integration of three spatial inputs with both ON and OFF responses. A minimal model in the ON pathway suggested that one input neuron with the opposite response polarity (here: OFF) on the trailing site of the T4 receptive field suffices to implement a passive mechanism of direction selectivity[20,37]. Voltage recordings in T5 showed that passive integration also occurs in the OFF pathway[21]. Another study that focused on T5 revealed that the T5 receptive field has both ON and OFF subunits that are contrast opponent, i.e., reversing the stimulus polarity at each receptive field location leads to a reversal in response polarity[7]. This contrast-opponent receptive field structure is used to produce direction selectivity through linear summation[7,17]. Thus, receiving both ON and OFF information at all input locations agrees with the recently reported linearity of T5 receptive fields. Whether or not the same structure applies to T4 in the ON pathway remains to be seen. Interestingly, a recent model of illusory motion perception in *Drosophila* required that the inhibitory input at the leading site of T4 (Mi4) responded linearly to convey inhibition to ON and disinhibition to OFF[44]. Analogously, the inhibitory input on the leading site of T5, CT1, provides inhibition to OFF and disinhibition to ON (Fig. 5)[11]. This suggests that T4 and T5 at least share a similar input mechanism on their trailing sites.

All OFF pathway Tm neurons were previously shown to respond positively to OFF stimuli[14,33,34,36]. Here, we showed that Tm9, Tm2, and CT1 also respond negatively to ON stimuli. All

Tm neurons are cholinergic and considered to provide excitatory input to T5, whereas CT1 neurons are GABAergic and likely inhibitory to T5[13,45] (Fig. 7h, i). The OFF excitation from Tm neurons and OFF inhibition from CT1 neurons (Fig. 7i) could be enough to describe the nonlinear summation mechanism observed for dark bars[21]. With the newly identified ON receptive field properties, all three dendritic input sites thus also receive ON information: from Tm9 on the distal part of the dendrite, from Tm2 centrally, and via CT1 at the basal site (Fig. 1). Because all subunits respond positively to OFF and negatively to ON, T5 receives contrast-opponent inputs across its receptive field.

Thanks to the extensive computational, behavioral, anatomical, genetic, and physiological characterization of the fly visual system, the substrate of this important neural computation is now well understood. In particular, measuring physiological response properties of the involved neurons reveals how a linear mechanism of motion computation can be implemented. This exemplifies how a thorough understanding of the neuronal circuit elements is necessary to unravel the implementation of the diverse computations performed by the brain.

**Negative ON responses in OFF pathway neurons**. Our work characterizes the response properties of the main input neurons to direction-selective T5 cells. The OFF-pathway neurons Tm9, Tm2, and CT1 display visual responses to ON stimuli and can thus implement a contrast-opponent mechanism for computing motion by providing the hitherto unknown ON input to direction-selective T5 neurons of the OFF pathway. Earlier difficulty in observing ON responses may be due to the relative nature of calcium imaging. While electrical recordings have a concrete reference voltage, calcium imaging is influenced by baseline fluorescence, which is affected by various factors, including sensor expression level and basal cellular calcium, and largely determines the dynamic range of detectable changes. Raising this baseline allowed the detection of response decrements using calcium imaging in previous studies looking at color and motion opponency[22,46]. Here, our stimulus design favored the identification of calcium decrements in response to ON stimuli by raising the baseline calcium signal of OFF Tm neurons with a dark background. Interestingly, T5 did not show significant negative calcium responses to ON bars, even when using a dark background. This is consistent with the notion of a rectifying nonlinearity of voltage to calcium in T5[7,17]. While direction selectivity exists in T5 voltage responses, it is considerably higher in T5 calcium responses. This conundrum of

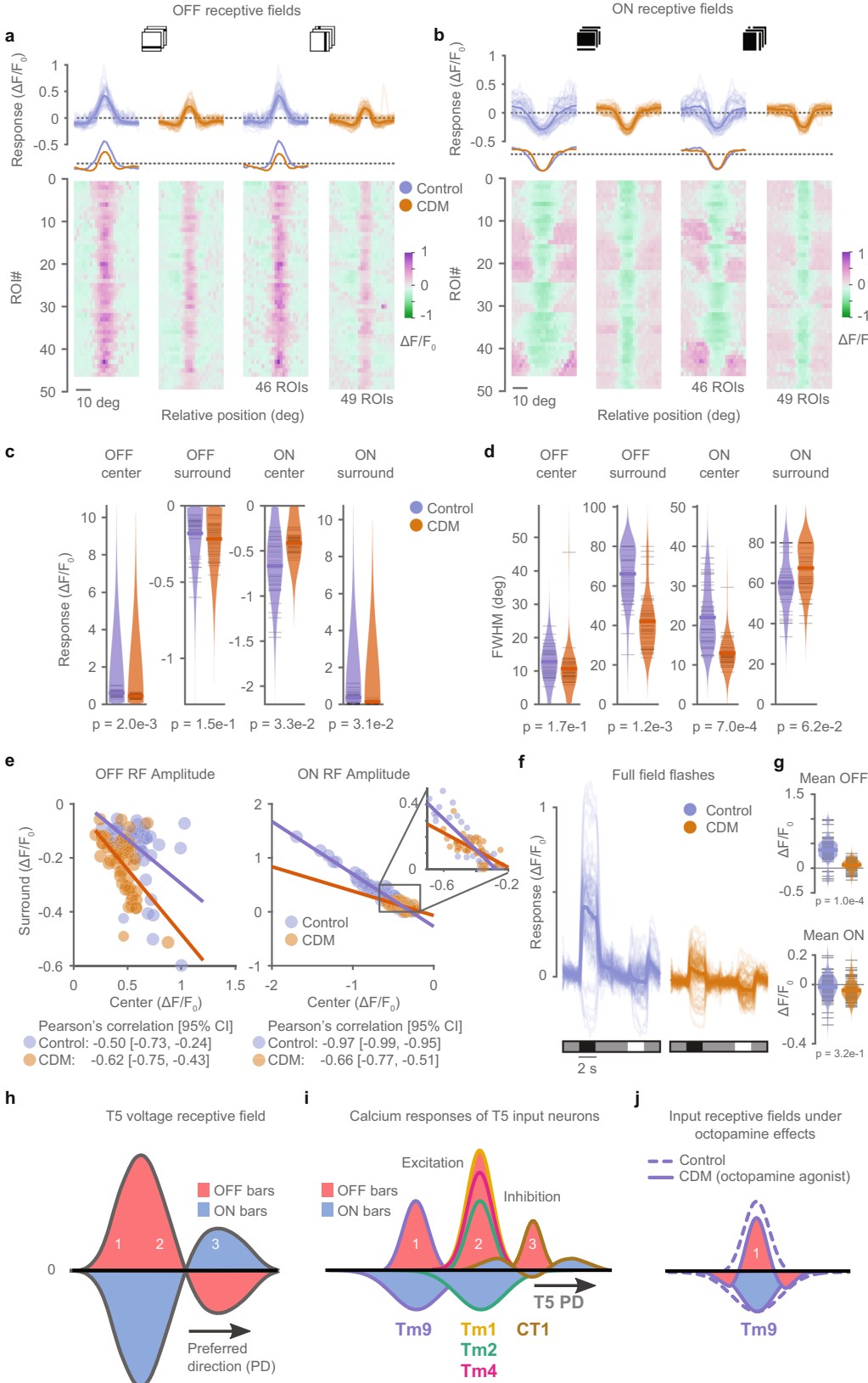

distinguishing linear from nonlinear processing steps was also addressed in studies of cortical simple cells. Whereas direction selectivity of spiking responses is nonlinear, membrane voltage integration is a linear process that is then followed by a nonlinear spiking threshold that amplifies the initial voltage selectivity[47].

**Harnessing variability through simultaneous measurements**. Our work shows that neuronal responses of an identified cell type can vary between cells within the visual system. To link Tm9 and T5 properties, simultaneous measurements of their responses are crucial because of the variable nature of the neural response

**Fig. 7 Chlordimeform (CDM) sharpens Tm9 receptive fields. a, b** OFF (**a**) and ON (**b**) receptive fields obtained from in vivo two-photon calcium imaging of responses to horizontal and vertical, dark and bright bars from Tm9 neurons expressing GCaMP6f, without (blue) and with CDM (orange). Traces below controls are the mean curves in **a**. Same neurons for ON and OFF: control $n = 46$, CDM $n = 49$, shown as thin lines. Only neurons with response quality index above 0.5 to all stimuli were included. **c, d** Quantification of the receptive field center and surround amplitudes (**c**), and full width at half maximum (FWHM) (**d**), obtained from a difference-of-Gaussians fit to data in **a**, **b**. Each point is the average of the fit amplitude (**c**) or fit FWHM (**d**) of a neuron across orientations. Two-tailed permutation tests comparing controls and CDM, p-values are indicated. **e** Comparison of receptive field amplitude of center and surround components for ON and OFF receptive fields shown in **c**. Lines indicate the least-square line fitting the data. Larger slopes indicate stronger center-surround interaction. Inset in the right shows an augmented version of the box. **f** Tm9 responses to full field flashes to ON and OFF stimuli (2 s) interleaved by an intermediate stimulus background (4 s), without (blue) and with (orange) CDM. Control $n = 70$, CDM $n = 86$. Few cells show a decrease in calcium signal in response to OFF, likely because their receptive fields are positioned outside the boundaries of the screen (see also [14]). **g** Distribution of the mean response amplitude of Tm9 during the presentation of the OFF and ON components from data in **f**. p-values were computed from two-tailed permutation tests. **h–i** Physiological basis of T5 direction-selective receptive fields. **h** Voltage receptive fields of T5 axon terminals showing ON and OFF subunits[7] (same as Fig. 1c). **i** Schematic showing receptive fields recorded using ON and OFF bars (Fig. 4, 5). The receptive field properties of Tm9, Tm1, Tm2, Tm4, and CT1 can explain the receptive field structure of T5 in **h**. **j** Schematic showing the sharpening of Tm9 ON and OFF receptive fields upon application of CDM, an octopamine agonist mimicking active behavioral states. Source data are provided as a Source Data file.

properties. So far, response properties of one cell type have been considered to be stereotypic, and in general, responses of different cells of one cell type are averaged to describe a cell type's properties[10,14,32,34,38,48,49]. In general, variability can be used to decorrelate information in the circuit and increase encoding efficiency by reducing the redundancy of information transmitted by a neuronal population[50]. For example, visual signals from photoreceptors are decorrelated after passing through the retina, LGN, and V1, and this decorrelation is accompanied by increased encoding efficiency[51–53].

Variability within a single cell type can emerge from intrinsic differences in the transcriptional landscape. Current advances in single-cell sequencing technologies are opening up avenues to study variability within neurons that are strictly thought to belong to one cell type by anatomical and genetic means. Recent studies identified dorsal and ventral Tm9 subtypes[54,55], which could contribute to the large variability of Tm9 ON receptive field sizes reported here (Fig. 2). Furthermore, variations in synaptic connectivity within a cell type might also account for the physiological diversity of Tm9 responses. Detailed information about connectivity is often limited to information on single or only a few cells from individual flies. It is very intriguing that Tm9 has been shown to be the most variable when comparing connectivity between seven neighboring columns in the medulla[56].

**ON and OFF inputs for direction selectivity persist across behavioral states.** Sensory systems have evolved to match the requirements of the surrounding environment. Thus, the mechanisms for motion detection must be able to accommodate the wide distribution of natural stimuli in the world, and the variation of the stimulus distribution caused by the behavioral state of the organism. An active behavioral state, such as walking or flying, comes with higher relative speeds of visual motion cues that the visual system needs to handle, and is known to modulate the temporal frequency tuning of visually guided behaviors[39,41,57]. Octopamine signaling modulates the temporal response properties of neurons at each stage of visual processing from the lamina[58], to T4/T5 neurons and their medulla input neurons[32,59], and then to wide field neurons downstream of T4/T5[40,43]. In vertebrates, locomotion modulates visual responses from the early visual system[60] to V1 cortical neurons[61].

In *Drosophila*, T5 neurons shift their temporal tuning to higher frequencies in the presence of CDM[32], which can now be explained by T5 receiving less OFF inputs from its slowest presynaptic partner Tm9, while maintaining similar suppression of Tm9 ON inputs. Furthermore, octopamine does not only act in the temporal domain[32,59], but our work shows that it also ensures spatially precise sampling (Fig. 7j). Mimicking an active state in

the fly led to a spatial sharpening of Tm9 ON and OFF receptive fields, and a reduction in amplitude for OFF receptive field centers, which enhanced center-surround interactions and consequently reduced responses to global stimuli. Together, these properties could ensure spatially and temporally precise sampling when the fly is flying or walking. Thus, the physiological basis for direction selectivity described here can perform across behavioral states by changing the neuronal properties to accommodate the varying stimulus statistics.

**Universality of contrast-opponent mechanisms for direction selectivity.** Our study in *Drosophila* reveals the neural substrate underlying a linear mechanism of direction selectivity, a solution found by neural circuits in a range of species. Linear mechanisms were first proposed to explain the motion responses of cortical simple cells[24], and primate motion perception[62]. Simple cells in the primary visual cortex receive thousands of synaptic inputs coming from thalamic neurons in the LGN, as well as from other cortical neurons. Most LGN neurons exhibit receptive fields with a classical center-surround structure and a push-pull organization[63], i.e., stimuli of opposite contrast lead to synaptic inputs of opposite polarity. In the Hubel and Wiesel view, simple cells are direction selective because they receive inputs from several LGN neurons, leading to receptive fields with two elongated ON and OFF subunits[64]. Recent evidence shows a V1 neuron receives spatially offset inputs from excitatory and inhibitory neuron populations in agreement with its direction selectivity[65]. In carnivores and primates, the simple cell subunits also have a push–pull structure[27–29]. Similarly, T5 neurons show a contrast-opponent organization (Fig. 1c), as do the Tm9, Tm2, and CT1 input neurons, although direct measurements of excitatory and inhibitory input currents would be needed to confirm a push–pull property. Our findings highlight the conservation of contrast opponency across the visual hierarchy in organisms as different as cat and *Drosophila*. Because T5 receives orders of magnitude fewer input synapses than cortical cells, and only a few cell types constitute the large majority of input synapses[13,25], we were able to map the detailed circuit structure, as well as the physiological contribution of identified cell types for a neural computation that is present in many organisms. Thus, the fly offers the opportunity to understand the minimal motifs behind mechanisms of direction selectivity.

## Methods

**Fly husbandry and preparation**. *Drosophila melanogaster* were raised on molasses-based food at 25 °C and 55% humidity, on a 12:12 h light-dark cycle. Female flies, 1–7 days after eclosion, were used for all experiments.

To open an optical window to their brain, flies were first immobilized in an empty vial by cooling on ice. Flies were inserted in a sheet of stainless steel foil, such that the thorax protruded but the rest of the body, particularly the eyes, remained below the foil. The fly was fixed to the foil by UV-cured glue (Bondic) applied to the thorax and the left portion of the head. To expose the right optic lobe for imaging from above, the cuticle, fat bodies, and trachea were removed using razor blades and forceps under ice-cold, low-calcium saline. Following dissection, low-calcium saline was exchanged with calcium saline at room temperature. Saline for calcium imaging was composed of 103 mM NaCl, 3 mM KCl, 5 mM TES, 1 mM $NaH_2PO_4$, 4 mM $MgCl_2$, 1.5 mM $CaCl_2$, 10 mM trehalose, 10 mM glucose, 7 mM sucrose, and 26 mM $NaHCO_3$ (no calcium, no sugars for low-calcium saline). The solution was bubbled with carbogen (95% $O_2$, 5% $CO_2$) and continuously perfused the fly brain (60–100 mL/h).

### Genotypes.

- $w/w+; Tm9^{24C08}\text{-}LexA,lexAop\text{-}GCaMP6f/+;+/+$
- $w+/w-;Tm9^{24C08}\text{-}LexA,lexAop\text{-}GCaMP6f/Tm4\text{-}split\text{-}Gal4\text{-}AD;UAS\text{-}jRGECO1a/Tm4\text{-}split\text{-}Gal4\text{-}DBD$
- $w+/w-;Tm1\text{-}split\text{-}Gal4\text{-}AD/+;Tm1\text{-}split\text{-}Gal4\text{-}DBD/UAS\text{-}GCaMP6f$
- $w+/w-;Tm2\text{-}split\text{-}Gal4\text{-}AD/+;Tm2\text{-}split\text{-}Gal4\text{-}DBD/UAS\text{-}GCaMP6f$
- $w+/w-;Tm4\text{-}split\text{-}Gal4\text{-}AD/+;Tm4\text{-}split\text{-}Gal4\text{-}DBD/UAS\text{-}GCaMP6f$
- $w+/w-;CT1^{R65E11}\text{-}Gal4\text{-}AD/UAS\text{-}GCaMP6f;CT1^{R20C09}\text{-}Gal4\text{-}DBD/UAS\text{-}GCaMP6f$
- $w+/w+;T4/T5^{R59E08}\text{-}LexA,lexAop\text{-}GCaMP6f;UAS\text{-}jRGECO1a/Tm9^{24C08}\text{-}Gal4$

**Imaging.** In vivo calcium signals were recorded using a Bruker Investigator two-photon microscope (Bruker, Madison, WI, USA) coupled to a tunable laser (Spectraphysics Insight DS+) with an additional output fixed at 1040 nm. The microscope was equipped with a 25×/1.1 water-immersion objective (Nikon, Minato, Japan). Laser excitation was tuned to 920 nm for GCaMP6f only measurements, and to 935 nm for dual-imaging experiments with both $T5\gg GCaMP6f$ and $Tm9\gg jRGECO1a$. For dual imaging of $Tm9\gg GCaMP6f$ and $Tm4\gg jRGECO1a$, the main laser output was tuned to 920 nm, and 1040 nm excitation was additionally delivered. Typically less than 20 mW of excitation was delivered to the specimen, measured at the objective.

Emitted light was sent through a SP680 short-pass filter, a 560 lpxr dichroic filter (which separated green and red emission) and either a 525/70 or a 595/50 emission filter. PMT gain was set to 855 V for both channels. The microscope was controlled with the PrairieView (5.4) software. Imaging rate was 8–12 Hz for most recordings, images of approx. 90 × 256 pixels were recorded, using an optical zoom of 8× to 10×.

### Visual stimulation

*Setup.* Stimuli were programmed in C++ using OpenGL, and projected using a LightCrafter 4500 DLP (Texas Instruments, Dallas, TX, USA) with only blue LED illumination. Stimulus light was attenuated with a 482/18 bandpass and ND1 filters, before reaching a rear projection screen. The screen measured 8 cm × 8 cm, subtending about 60° × 60° (azimuth × elevation) of the right visual field of the fly. Stimuli were displayed at 6-bit pixel depth and at a frame rate of 300 Hz, but the stimulus frame was updated at 100 Hz. The stimulus frame parameters, including timestamps, were saved to disk together with the imaging data frame timestamps to allow for stimulus-imaging time alignment during data analysis. The stimulus and data acquisition computers were linked via a NI-DAQ USB-6211 device (National Instruments).

*ON/OFF bars.* To measure spatial receptive fields, stimuli consisted of 5° bars. ON and OFF bars were 100% Michelson contrast: bright bars on a dark background and vice versa. Individual bar positions covered the screen in 2° shifts. For each trial, bar positions were shuffled to be presented in a pseudo-random order. A single bar was flashed at each position for 1 s, with a 1 s inter-stimulus interval at which background was shown. In total, bars of a single polarity and orientation (horizontal or vertical) were presented for four to five trials each.

*Ternary noise.* To measure spatiotemporal receptive fields, stimuli consisted of ternary noise. Each stimulus frame consisted of 12 bars tiling the screen either horizontally or vertically. The bars were 5° wide. For each frame, each of the 12 bars was assigned a black, white, or gray contrast value with equal probability, and independent from the other bars. The contrast of the bars was updated every 100 ms. A 3 s gray background was shown before starting the ternary noise stimulus.

*Sine wave gratings.* To measure spatiotemporal frequency tuning, stimuli consisted of moving sine wave gratings. The gratings moved in the back-to-front direction, and had 100% Michelson contrast. Spatial and temporal frequency was drawn from a two-dimensional parameter grid. The grid was constructed using the spatial wavelengths 2°, 3.5°, 6°, 10°, 18°, and 32° per cycle, and temporal frequencies 0.25, 0.5, 1.25, 3, 8, and 16 Hz . In total, 36 gratings were shown for a single stimulus trial in a pseudo-random order. Gratings were presented for 4 s with 4 s gray background between gratings. The stimulus sequence was presented for five trials.

### Data analysis

*Data processing.* Imaging time series were registered to compensate for within-plane motion of the specimen. Image registration used either cross-correlation alignment or RASL (robust alignment by sparse and low-rank decomposition)[66]. To align across the time series of the same recording session, each time series was first independently registered to obtain a within-time-series registration. Then, the mean of each independently registered time series was used to register all time series across the recording by applying the obtained global shift to all frames of the within-time-series registered frames.

Fluorescence time series $F(t)$ were high-pass filtered with a cutoff period of ≈0.1 Hz (or 150 data frames). ROIs were selected manually following the stereotypical shapes of the recorded neuron types. Pixels were averaged within each ROI. The time series was normalized as $\frac{\Delta F}{F_0} = \frac{(F-F_0)}{F_0}$, where the baseline fluorescene $F_0$ was chosen as the average fluorescence during all presentation of the background stimulus. To avoid strong signal fluctuations for recordings with baseline signal close to 0, the mean fluorescence of the full time series $\overline{F}$ was added to the denominator $\frac{\Delta F}{F_0} = \frac{(F-F_0)}{(F_0+\overline{F})}$.

The time of the fluorescence signal was aligned to the time of stimulus presentation before trial averaging. To average signals acquired at slightly different frame rates, all time series were interpolated to a common frequency of 10 Hz. The response variability of the fluorescence time traces across stimulus repetitions was quantified using a response quality index[67]. This index is the signal-to-noise ratio given by $Q_i = \frac{\text{var}[\langle F \rangle_{trials}]_{time}}{\langle \text{var}[F]_{time}\rangle_{trials}}$, which is the variance across time of the response trial average divided by the trial average of the variances across time. For identical responses across trials this index equals one, while for completely random responses with fixed variance the index is inversely proportional to the number of trials. A threshold value of 0.5 for the response quality index was used to select responding neurons independently of the stimulus structure (Figs. 2, 4, 5, and 7). For dual color imaging, we applied a lower response quality index of 0.3 due to a lower signal-to-noise ratio of the jRGECO1a calcium indicator (Fig. 6).

*Tuning curves. Spatial receptive fields.* To quantify the receptive field position, width and amplitude, we obtained a tuning curve as follows. The 1 s response traces to (shuffled) bar positions were averaged across trials and sorted by spatial coordinates (Supplementary Figs. 2 and 3). Taking the maximum absolute value of the response to each 1 s bar presentation resulted in a tuning curve. This tuning curve (example Fig. 4b–e) was then fitted to a single Gaussian curve of the form $f(x) = A \exp\left(\frac{-(x-x_0)^2}{w^2}\right)$. The amplitude is given by $A$, the position by $x_0$, and the width by $w$. The full width at half maximum (FWHM) is then calculated as $\text{FWHM} = 2w\sqrt{\log 2}$. To quantify center-surround receptive fields, the same procedure was followed but the function fitted to the tuning curve was a difference-of-Gaussians of the form $f(x) = A_c \exp\left(\frac{-(x-x_0)^2}{w_c^2}\right) - A_s \exp\left(\frac{-(x-x_0)^2}{w_s^2}\right)$. Both center and surround are constrained to the same spatial location $x_0$, but allowed to have different amplitudes ($A_{c,s}$) and widths ($w_{c,s}$). We constrained the difference-of-Gaussians fitting algorithm to solutions with $A_c > A_s$, and $A_{c,s}$ amplitudes below twice the amplitude range of the original data.

Gaussian fits of Tm9 neurons were considered for quantification if their goodness-of-fit ($r^2$) was greater than 0.5 for OFF bars, and 0.2 for ON bars (Fig. 2). For data of different Tm neurons matched across stimuli neurons had to pass the same thresholds for both orientations (Figs. 4 and 5). For the same dataset of Fig. 4, all tuning curves without any selection criteria are displayed in Supplementary Fig. 4. For dual color Tm4 and Tm9 imaging, because Tm4 neurons did not respond to ON bars (Supplementary Fig. 3), a threshold of 0.8 was set for OFF responses of both Tm9 and Tm4 (Fig. 3). For data in Fig. 6, fits to responses to OFF bars were considered for $r^2 > 0.25$ for both Tm9 and T5. For Tm9 recordings with and without CDM, where both center and surround of the response was analyzed, difference-of-Gaussian fits were applied only for neurons passing the response quality threshold for all conditions (Fig. 7 and Supplementary Fig. 6).

*Spatiotemporal receptive fields* were obtained from responses to ternary noise using the fluorescence-weighted average stimulus. The spatiotemporal receptive fields (linear filters) are given by $k(x,\tau) = \frac{1}{(T-\tau)} \cdot \sum_{t=\tau}^{T} r(t)s(x,t-\tau)$, where $r(t)$ is the $\frac{\Delta F}{F_0}$ signal, $\tau$ is the time window of the stimulus average, and $s(x,t)$ is the value of stimulus at position $x$ and time $t$. The spatiotemporal receptive field was normalized to have a unit Euclidean norm (arbitrary units), and was further split into a spatial and temporal receptive field by taking one-dimensional slices along the respective dimensions from the point of maximum receptive field strength. The spatial receptive field was fitted to a Gaussian curve similarly to the bar receptive fields.

*Spatiotemporal frequency tuning* was calculated using the Fourier amplitude spectrum of the response for the frequency matching the temporal frequency of the grating. The tuning similarity across the simultaneously imaged neuron types was quantified as the Pearson correlation between the two-dimensional maps of each neuron pair.

**T5 dendrite model**. The model was coded in Python using the NEURON simulation software. The geometry of the fly eye was approximated by an hexagonal array consisting of a central point surrounded by six neighbors spaced by 5°. The points in the array defined the position of the receptive fields from the medulla input neurons Tm1, Tm2, Tm4, Tm9, and CT1.

The spatial receptive field of the neurons consisted of a difference-of-Gaussians. The temporal component was a low-pass filter for CT1 and Tm9 ($f_{LP}(t) = 2\tau^{-3/2}te^{-t/\tau}$), and a high-pass filter for Tm1, Tm2, and Tm4 ($f_{HP}(t) = 2\tau^{-3/2}(\tau - t)e^{-t/\tau}$)[37]. Parameters for the spatiotemporal receptive fields follow from a previous study[32]; parameters for Tm9 were transferred to CT1, because CT1 receives the majority of inputs from Tm9.

A visual stimulus consisting of a sinewave grating was first transformed using the spatiotemporal receptive field into an input current to the medulla input neurons. The current was then converted into a conductance change using synaptic dynamics with two time constants (alpha synapse) taken from literature[20]. This conductance was fed into a nonlinearity approximating the contrast selectivity of the neurons. The nonlinearity was either a linear function indicating ON and OFF responses (positive for OFF, negative for ON), or a rectified linear function indicating preference for only OFF responses (zero ON, positive for OFF). The conductance output after the nonlinearity was then offset by a fixed constant equal for all neurons, to prevent negative conductances in the range of contrasts used in the simulations, and keep them in the unit range [0, 1]. The resulting conductance is the input from each neuron to the T5 dendrite model, which is weighted proportionally to the synaptic connectivity given by recent connectomics data[13]. Inputs from Tm1, Tm2, Tm4, and Tm9 were excitatory with reversal potential at 0 mV, while inhibitory inputs from CT1 had a reversal potential of −70 mV. The amplitude of the input conductances was $2.49 \times 10^{-5}$ μS for excitation and $4.98 \times 10^{-4}$ μS for inhibition, reflecting the range of values in a previous study[20]. Our only free parameter is the inhibition to excitation ratio ($I/E$), which controls the relative strength of excitatory and inhibitory synapses, set to 55 for all simulations except the ones in Supplementary Fig. 1f–h. The T5 dendrite was simulated as a passive cable with 13.3 μm length and 0.2 μm diameter. Using parameters from a T4 dendrite model[20], the passive cable had 100 Ohm cm axial resistance, 1 μF cm$^{-2}$ membrane capacitance, $1.03 \times 10^{-4}$ S cm$^{-2}$ membrane conductance, and −65 mV leak potential. The T5 dendrite received inputs as follows: Tm9 inputs from the trailing column were given close to the dendrite tip; Tm1, Tm2, and Tm4 inputs from central column were given close to the dendrite center, and CT1 inputs from the leading column were given near the dendrite base. Finally, voltage responses of T5 were measured from the dendrite base.

To calculate direction selectivity, we simulated responses to gratings moving in 16 different directions for 6 s; the gratings had a temporal frequency of 1 Hz and a spatial frequency of 1/24 cycles per degree. We quantified the dendrite response as the component of the Fourier amplitude spectrum at the frequency of the stimulus. The response amplitude and the direction of the stimulus can be represented as a vector in polar coordinates. This way we measured the direction selectivity index (DSI) as the amplitude of the sum of the response vector across motion directions, divided by the sum of the response amplitudes. If all vectors respond equally in all directions, the vector average will be zero, indicating no direction selectivity, and if only one direction elicits responses, then the DSI will be one.

This model draws parameters from literature[20,32] and simulates a simple T5 dendrite constrained by the properties of the input neurons. While a recent model[20] fitted the parameters to match T4 data without any direct consideration of the properties of the input neurons, we expect the synaptic parameters to be a good approximation to the real biophysical properties of T5 dendrites yet to be characterized. To test the robustness of the model to the readout quantity, besides the first harmonic amplitude, we also quantified the maximum of the dendritic voltage response, as well as the mean of the dendritic calcium response (Supplementary Fig. 1c–e). The calcium response was obtained by squaring the positively-rectified voltage response ($[\Delta v]_+^2$)[17,37]. We tested how the direction selectivity is influenced by our free parameter, the $I/E$ ratio (Supplementary Fig. 1f–h), and temporal frequency (Supplementary Fig. 1i–k).

**Statistical analysis**. Nested permutation tests were used for statistical comparisons in Figs. 2 and 7 to account for the potential correlations of neurons within the same fly. This was done by permuting all neurons from the same fly together rather than permuting neurons independently from each other. In Fig. 2f, non-nested permutations were used, since the number of neurons per fly was low to contribute to within fly variability. In Fig. 6f and Supplementary Fig. 5a, the Pearson's correlation and corresponding p-value were given by the corrcoef MATLAB command. And in Fig. 6h, the significance of the correlation between the spatiotemporal frequency tuning maps of pairs of overlapping Tm9 and T5 neurons was calculated as follows. The Tm9 and T5 pairs were randomly shuffled, the pairwise correlations were calculated, and the mean of this shuffled distribution was computed. The process was repeated 1e6 times resulting in a shuffled control distribution. A one-tailed p-value was calculated as the ratio of mean shuffled

correlations larger than the original mean correlation to the total number of shuffles (10⁶). A p-value of $p < 10^{-6}$ was obtained, since no mean shuffle was ever higher than the original mean. In Fig. 7e, the Pearson's correlations and their 95%-confidence intervals were computed by 1000 bootstrap runs.

**Reporting summary**. Further information on research design is available in the Nature Research Reporting Summary linked to this article.

## Data availability
Source data are provided with this paper as a Source Data file. The minimal dataset and analysis code to generate the figures in this study have been deposited in the G-NODE database under accession code 10.12751/g-node.qeeyfz [https://gin.g-node.org/GRamosT/off-motion-receptive-fields][68].

## Software availability
Software for the model is deposited at https://github.com/silieslab/RamosTraslosheros_Silies_2021[69].

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

## Acknowledgements

We thank Carlotta Martelli, Julijana Gjorgjieva, and members of the Silies lab for discussions and critical comments on the manuscript. We are grateful to Aljoscha Nern and Michael Reiser for sharing fly stocks. This project has received funding from the European Research Council (ERC) under the European Union's Horizon 2020 research and innovation programme (grant agreement No 716512).

## Author contributions

G.R.-T. and M.S. conceived the experiments. G.R.-T. conceived the model, performed the experiments and analyzed the data. G.R.-T. and M.S. wrote the paper.

## Funding

## Competing interests

The authors declare no competing interests.
