## [Peer Review File · Nature Communications]

Reviewers' Comments:

Reviewer #1:

Remarks to the Author:

Ramos-Traslasheros & Silies present results showing how the response properties Tm9 neurons influence responses of T5 direction-selective neurons. They measure calcium signals in Tm9 (and other input neurons to T5) in response to a variety of stimuli, show a correlation between Tm9 responses and downstream T5 responses, and show how Tm9 responses are modulated by the octopamine agonist CDM. They relate the response properties of Tm9 to the direction-selective computations in T5 by a numerical model.

This work demonstrates that both Tm9 and Tm2 respond to contrast increments (ON stimuli) by hyperpolarizing, in contradiction to previous studies that modeled them as only depolarizing to OFF stimuli (Serbe et al.). Tm9 neurons have receptive fields of different sizes in response to ON vs. OFF stimuli, which is an interesting result and further resolves an apparent discrepancy in the field (Fisher et al., 2015 vs. Arenz et al., 2016). It is important because it also serves as an excellent example (and warning) for how stimulus protocols can effect measured properties. The data in this paper is high quality, but there are significant issues that should be addressed in its presentation.

Major

1) There were two issues with terminology that confused this reviewer.

a. First, "linear computation of direction selectivity", referenced throughout, seems like a contradiction in terms. It is widely recognized that direction-selectivity requires a nonlinearity. This is said explicitly both in Adelson & Bergen's seminal paper on the motion energy model and in Egelhaaf & Borst's 1989 review. What is described here as 'linear direction selectivity' is the linear processing upstream of a nonlinearity that generates direction-selective signals. This needs to be made clear in the paper, starting in the title, abstract, and introduction.

b. As a related point, from the first sentence of the abstract and the introduction, this manuscript takes as canon that there is a linear transformation from stimulus to voltage in Dmel DS neurons. This should be tempered when introduced: though Wienecke argued for this, the two Gruntman papers spent substantial time on the -nonlinear- null-direction suppression aspects of their voltage measurements, caused by current shunting. Similarly, Badwan suggested that the nonlinear interactions between conductances might be responsible for phenomena that were not possible with a purely linear mechanism.

c. The second terminological issue that confused me was the use of ON and OFF. In the original usage of these terms (Kuffler 1952), units were classified as ON or OFF by whether they fired to contrast increments or decrements. ON and OFF neurons later came to be neurons that depolarized to increments or decrements; the definition is independent of rectification, as for example applied to quite linear bipolar cells or to tonically firing ganglion cells (Enroth-Cugel & Robson, 1966; Werblin & Dowling 1969). It was very disconcerting for the hyperpolarization of an OFF cell (Tm9) to be called an 'ON response' (lines 109, 111, 135, 136, and many other places). As used in the manuscript, a perfectly linear cell would have both an ON response and an OFF response; this leads to confusion. More generally, there are ON/OFF stimuli, ON/OFF edges, ON/OFF cells, and depolarizing and hyperpolarizing responses in cells to these stimuli. These should be very explicitly defined and referred to so as not to lead to confusion based on prior definitions in the field.

2) The authors say several times, including in a discussion section, that their data imply a push-pull mechanism. I do not think the data here support this claim. Push-pull amplifiers use two transistors, one to pull voltage up, one to pull it down. In neural circuits, push-pull is canonically used to describe neurons that are excited by inputs of one polarity and inhibited by inputs of the opposite polarity – that is, the excitatory and inhibitory inputs are critical to this description, not merely depolarization or hyperpolarization. What is described here is instead more like "push-push less", since the idea is that Tm9 is excitatory with some tonic activity that can increase or decrease with contrast decrements or increments, respectively. The reduction in excitation from Tm9 onto T5 due on an ON stimulus causes T5 to hyperpolarize relative to baseline. That model for neural interactions would not be a push-pull model. For instance, in vertebrate retina, photoreceptors release glutamate tonically, and the release can be modulated up or down; the synapse from PRs onto BPCs is not considered to be push-pull. Moreover, if push-pull applied whenever a neuron depolarizes to one polarity of stimulus and hyperpolarizes to another, then

every linear visual neuron would be described as 'push-pull'.

3) As presented, there appear to be two important conceptual issues with the model.

a. First, the authors say multiple times that their model integrates neural inputs linearly. But their numerical Neuron model should not be linear in its conductance inputs: (a) reversal potentials for each synapse mean postsynaptic voltage saturates with increasing conductance and (b) shunting in currents can occur between synapses, so that conductances interact. This sort of passive membrane model is linear for injected currents, but not for postsynaptic conductances, which saturate and interact nonlinearly. The model does not appear to do what the authors say it does. In fact, the model used here seems very close to the nonlinear interactions used to model data in Gruntman et al. and in Zavetone-Veth et al.

b. Second, the description of the model in the methods says that negative conductances are possible when there's no rectification on the inputs to T5 (lines 504-507). Negative conductances are clearly non-physical, so should not be present in any biophysical model. In the case of modeling a nonrectified response, some other means must be found to keep the conductance non-negative and linear (perhaps within a given range of input contrasts). If offsets are added, then comparisons need to be made with similarly offset rectified inputs.

4) The DSI in figure 1 was measured by finding the amplitude of the response component at the temporal frequency of the input grating. This is a nonlinearity applied to the voltage trace, but it's not clear what kind of biophysical mechanism could perform this operation by isolating a specific frequency. The nonlinearity chosen here determines the direction-selective properties of the response if upstream processing is mostly or entirely linear, so it's important to choose a nonlinearity that could be plausibly (or cartoonishly) implemented in a biological system. The authors should justify the specific nonlinearity they choose.

5) The correlation between Tm9 and T5 receptive field sizes is interesting. As the authors note, each Tm9 is presynaptic to 4 T5 cells, and I agree with the argument that their oriented stimulus choice removes two of these from consideration. However, since the authors record in the T5 dendrites and each T5 dendrite spans 3 columns, aren't the recordings in a single column really from up to 6 different T5 cells (say 3 each from LP layers a and b)? I believe this is true, though it may depend on the orientation of the recording plane. S1e doesn't really address this point because it looks at correlation in RF center over many ROIs. Really, one might want to know the distribution of RF center differences, which frequently look like they're off by 5 degrees. I'm not sure one even expects connected Tm9 and T5 cells to have peak OFF bar responses at the same position, does one? If the authors agree that connecting Tm9 to T5 responses is nontrivial here, then this section could use substantially more explanation and caveats. It would also help to know more about the variability here: was it within fly or between fly? Was it correlated or uncorrelated with position in lobula, or with position on the screen?

6) The last section title of the results says that behavioral state modulates Tm9 response structure. To make this claim, the authors have added a global octopamine receptor agonist while imaging. Thus, the behavioral state claim is too strong and should be softened, since the authors could have run alternative experiments to make this claim directly (as in Strother et al. or Chiappe et al.). (As an aside: Is this CDM result consistent with Arenz's results? I recall they also seemed to find deeper inhibitory lobes with CDM present. The authors might remark on similarities.)

Minor

1) 'Linear' is used in many instances, and it was not always clear what the linear is in reference to. For example, the authors discuss linear summation of inputs in T5, linear responses of T5 to stimuli, linear transmission from Tm9 to T5, and linearity of responses in Tm9. There are several places, especially related to T5, where I found it ambiguous whether the authors meant linear in space or linear responses to localized stimuli or linearity of inputs to T5.

2) Statistics: I believe the authors treat individual ROIs as statistically independent. It seems as though there are likely to be correlations within flies, though, so is this warranted? The low p-values in Figure 6c, for instance, could result from treating correlated data as independent.

3) Lines 66-70: Authors state here that ON inputs to T5 are required. Cells that transmit information about contrast increments are required, given Wienecke's result, but since OFF cells need not be rectified, they could suffice. This and several other comments below are examples of the terminology issue as noted above.

4) Line 71: ON inputs to T5. Responses to ON stimuli? ON cells? It's a bit ambiguous here, and elsewhere, and it would be profitable to eliminate this sort of ambiguity.

5) Line 74 and elsewhere: "OFF direction-selectivity" – not entirely clear how this use of ON and

OFF differs from others. T5 cells respond to moving dark edges, but as the authors point out, their linear receptive fields have positive and negative regions.

6) Line 102: "OFF-rectified inputs"? When 'rectified' is not specified, does that mean it's a potentially linear OFF input?

7) Line 117: ON inputs provided by linearizing an OFF excitatory input neuron? This is a confusing description. Line 119, too.

8) Lines 124, 172, 253: "ON input required" – figure 1 shows that linearity in Tm9 is sufficient to explain some measurements in T5, but not that it is required.

9) Line 125: reference to non-existent figure 7.

10) Line 136: Tm9 ON responses. Calling these "ON responses" is pretty confusing.

11) Question: something similar to this phenomenon was proposed for T4, which can surprisingly be depolarized by presenting dark stimuli offset from its RF center (Agrochao et al.). That paper suggested this might be disinhibition via Mi4, which is not entirely rectified and represents a possible pathway for OFF information to reach T4, beyond Mi9. Could the authors comment on whether these are related properties in the two pathways?

12) Line 152: The resolution of the RF size discrepancy in the literature is satisfying and nicely shown.

13) Figure 2d: could emphasize these are both spatial dimensions.

14) Figure 2d, 3rd column: the dark bar and flicker receptive field peaks appear offset in x. Is this remarkable?

15) Figures 3e and 2cf show same information, but different plots are used. Might make it easier on readers to use same presentation?

16) Line 169 (and elsewhere): why is this receptive field called "inhibitory"? The hyperpolarization of T5 to ON stimuli doesn't require an inhibitory ON input; as the authors show, it appears like to be an excitatory OFF cell responding at below baseline to ON stimuli – excitatory OFF and inhibitory ON look pretty similar in T5 when one cannot measure currents.

17) Fig 5d: p-value is cut off for me.

18) Fig 5c: why take the amplitude of the component at that temporal frequency? This is sort of the opposite of what Strother 2013 did for analyzing similar data – that paper evaluated the mean response over time to judge bandpassing properties.

19) Fig 6bd: why can't we see the significant sharpening of the ON-bar receptive field width in the mean traces in b? I'm surprised.

20) Fig 6e caption: "inset shows augmented version of dotted box"? What does this refer to? I can't figure it out.

21) Line 256, line 292 as well: Tm9 and 2 have inhibitory ON receptive field components – I don't think this paper provides evidence that these are inhibitory components, just that the cells hyperpolarize in response to ON stimuli.

22) Line 288: "close to complete" – it seems premature to declare this.

23) Line 292: These are not proven to be inhibitory. They're hyperpolarizing.

24) Line 293: Avoid editorializing on the importance of the work.

25) Line 296: "ON input to OFF direction-selective cells". I think this framing is confusing, given the use of this terminology in the field.

26) Line 354: swap 'primate' for 'human'?

27) The authors write that the variability in Tm9 RF size shows that it must be important for motion detection, but I think this is not necessarily true. Tm9 could feed into several downstream circuits, and could have properties related to non-DS functions. So to make this claim, one would need to demonstrate why this improves motion detection; its mere existence doesn't demonstrate that it is a circuit feature that improves motion detection.

28) Line 523: Here it says that the DSI is the sum of vectors divided by the sum of amplitudes; to get a scalar out, it must be the amplitude of the sum of the vectors, divided by the sum of the amplitudes.

29) In several figures (Fig. 2b, 4b-i), the number of ROIs for the light bar RF measurements are substantially smaller than for the dark bar RF measurements. But not in 6ab. This seems to be because only ROIs with significant responses are shown in Figs 2 and 4. Does this artificially increase the mean responses to ON stimuli? For instance, if one includes all ROIs that respond to either ON or OFF stimuli, is the ON-dependent hyperpolarization smaller, since one averages in many more responses of 0? Is this an important issue?

Reviewer #2:

Remarks to the Author:

In their manuscript entitled „The physiological basis for linear computation of direction selectivity“, Giordano Ramos-Traslosheros and Marion Silies use in vivo two photon calcium imaging to explain how linear summation in T5 cells leads to direction selectivity in the *Drosophila* OFF pathway during motion vision. By revealing unexpected ON-responses for both Tm2 and Tm9 neurons, the authors reveal that linear summation in the OFF pathway might indeed be more similar to the ON pathway (culminating on T4 cells), than previously known. Despite superficially sounding like ‘filling in the gaps’ his finding is very relevant in several ways. First, it shows that T4 and T5 inputs are most likely not using fundamentally different mechanisms, since they both combine ON and OFF inputs along the dendritic tree, resulting in a push-pull configuration. This was previously not known. Hence, this work provides important progress towards working out fundamental principles underlying motion vision. This is further highlighted by the multiple parallels to direction selectivity of mammalian cells, which are being worked out by the authors throughout the paper in an exemplary fashion. The data presented are of very high quality and to the most part very clear (see my comments below, for improvements). Importantly, the authors use GCamp6f as an indicator and successfully visualize hyperpolarization, thereby validating previous experiments from other groups that used either voltage imaging or electrophysiology. Besides providing such synergistic validation, the data provided here will be of great interest to anyone performing similar experiments/answering similar questions in a different experimental context. Furthermore, the authors use two color imaging in order to confirm the unusual properties of Tm9 (by directly comparing it to Tm4), and experiment that I would have requested, although knowing that it would be technically very challenging. It speaks for the high quality of the data presented here, that such an experiment is being provided.

I have quite a few comments regarding the Figures and how they could be improved, as well as a few comments on the text. If these points were to be addressed, then I recommend this manuscript for publication in Nature Communications.

Major points:

- The authors use the variability of Tm9 responses as a clever feature to demonstrate information flow from Tm9 to T5. In the process, some statements are exaggerated, in my opinion. For instance in lines 192/193, the authors speak of ‘biological relevance’. I find this term misplaced here. The upcoming experiment does not technically test for this, but merely investigates whether this feature is transmitted to the post-synaptic T5 cell. The same kind of sloppiness arises again in line 211, where the same variability is identified as being ‘important’ for downstream computations. Again, this is an over-interpretation of the data, since all it shows is that the variability is maintained. This might very well be important, especially since it makes sense computationally, but it hardly proof supporting such a strong statement. Finally, in line 259, the authors identify the tight coupling between Tm9 and T5 as being ‘critical’ for motion computation. Again, this is over-stated. The data is in line with the model, the response properties of the cells correlate, but is no proof for Tm9 being critical. These sentences should all be softened.

- Line 304: The authors state that T5 do not manifest Calcium responses to ON bars. Is that really true? When I clinch my eyes, I seem to see responses in Figure S1. Is there not even a trend? Can the authors comment on that? Regardless, I recommend moving the data from supplemental Figure S1 into Figure 5 (see below), since it is being discussed in detail, so it is weird not to be shown in the main Figures.

- Colors used in the Figures: I find it very confusing when purple and green are being used within the same panel, while encoding two different things, i.e. showing traces of two different cell types like Tm9 and T5, while at the same time encoding depolarization versus hyperpolarization for both cell types. This needs to be changed. I recommend using light blue and light orange for the traces, while maintaining the color code for activation and inhibition, or vice versa. Maybe changing the color code of the latter makes even more sense, since purple and green are also used when labeling the cell types using immune histochemistry.

- As stated above, the data from supplemental Figure S1 should be incorporated into Figure 5, since it is quite relevant and is being discussed rather prominently in the context of Figure 5.

- Figure 6F: Can the authors comment on the inverted response traces of quite a few individual Tm9 cells when stimulated with OFF and ON stimuli? Maybe I am not on top of what is known about Tm9, but some of these cells seem to hyperpolarize in response to OFF and depolarize in response to ON. How come?

- I think a final, summarizing Model figure linking back to Figure 1A would be very helpful, summarizing the ON/OFF inputs along the dendrite, and maybe even the octopaminergic modulation.

Minor points:

- Line 16: "...is passed ON to..."

- Line 31: Wouldn't "...AT LEAST three synapses away from photoreceptors..." be more accurate? There is always direct and indirect pathways, no?

- Line 36: Should electrophysiology from previous studies also be listed here, in addition to voltage imaging?

- Line 166: Rather than starting the sentence with 'because' I would start it with 'since'.

- Line 169/170. It would be more accurate to speak of Tm9 being a very suitable candidate for providing ON input onto T5

- Line 199: Rather than starting the sentence with 'because' I would start it with 'since'.

- Figure 1A: Please move the explanation of excitatory and inhibitory synapse arrows somewhere else (between the T4 and T5 dendrites?), since the way it is depicted now, their current position erroneously insinuates that they are homologous to the PD, in the OFF pathway.

- Figure 1G: Please move the previously published voltage imaging traces BELOW the outputs of the three models, simply because it is important to understand which traces belong to which model. This is most easily done by placing them right underneath each other.

- Figure 3B-D: I would find it more consistent if all Tm4 data was placed in the left column, and all the Tm9 data was placed in the right column. As it is now, cell type specific data is presented in columns (B) and rows (C,D) which is rather confusing.

Reviewer #3:

Remarks to the Author:

In this work, Ramos-Traslosheros and Silies study the mechanisms of direction selectivity (DS) in the T5 neurons in the fly. First, they construct a model of T5 dendrite receiving its known presynaptic inputs (Tm9 near the tip, Tm1, Tm2, Tm4 in the middle and inhibitory CT1 at the base of the dendrite). They stimulate the model with drifting gratings and observe the resulting DS. They show that the best DS is achieved with linear (non-rectifying) Tm9, Tm2 and CT1 cells vs the alternatives of all rectifying presynaptic cells or just rectifying Tm9 (figure 1). Second, to examine the level of rectification in presynaptic cells they express calcium reporters and map the receptive field (RF) of the cell with vertical and horizontal bars (figures 2-4). They find that Tm9 responds well to light bars with reduced calcium signals, irrespective of bar orientation (Figure 2e). Additionally, the spatial extent of RF size is much larger for ON bars or white noise (Figure 2f). Using a combination of red and green calcium indicators, the authors then compare the RF properties of Tm4 and Tm9 cells. Using such 'paired' recordings they confirm that Tm4 is a rectifying OFF cell. Its Rf size matches the size of the Tm9 OFF component (Figure 3). In figure 4 they combine RF analysis for Tm1, Tm2, Tm4 and Tm9 on a single panel and show that Tm2 and

Tm9 are the only cells with ON RF components and these components extend further in space than the OFF RFs of all cells. Next, the authors again use double recordings to measure the spatiotemporal characteristics of the Tm9-T5 cells with similar RF locations (supp figure) and reveal a Pearson correlation of about 0.4 between the widths of the RFs and highly similar responses to gratings (Figure 5). Last, CDM application to activate octopamine receptors reduced the amplitude of the responses in Tm9 and the size of the ON center component of the RF. I find the work to be interesting, but I do not think that the experiments and the framing of the paper as presented in the current format warrant the conclusions reached by the authors. Specifically:

1. First, the motivation for the study and the title. What do the authors mean by 'linear computation of DS'? As I understand it, the main point of the study is to understand the contrast responses of T5 presynaptic neurons. In the text I could see two different concepts of linearity (1) non rectifying contrast responses (lines 41-43) (2) previous work (notably Gruntman et al 2018, 2019, Wienecke et al 2018) showed that the computation of DS is performed by a linear postsynaptic integration over presynaptic units (lines 35-40). The authors seem to assume that (1) is required for (2), but why is that so? Would the results of the present study be invalidated if it turns out that Reichardt and Hassenstein were right all along? A much better motivation for this study would have been an exploration of contrast dependency of DS in T5 neurons, as this is what the model proposed by the authors (figure 1) shows. At the very least, the manuscript requires a major rewrite to address this point.

2. Line 88: The DS effect of negative ON excitation is not equal to an ON inhibitory input, as is clearly shown by Gruntman et al 2018.

3. The model (Figure 1)

3.1. While the model examines voltage attenuation in a T5 dendrite, the model itself is just a 13.3 um long, 0.2 um wide cable. From a biophysical standpoint, there is no significance to the spatial location of inputs in this tiny patch of membrane. The model doesn't approximate the actual voltage propagation in a T5 dendrite, which is highly branched and has one end that is leading to release sites while the other end gets to the soma. If the authors consider the spatial arrangement to be essential and want to consider dendritic voltage attenuation (Rall's model of DS) they should at least mimic this morphology.

3.2. Is there a reason to use traces from Wienecke et al 2018 which were based on voltage-sensitive dyes? Gruntman et al 2019 performed direct recordings from T5 neurons.

4. Using conductance-based models, Gruntman et al 2019 demonstrated that reduced excitation could not fully replicate the voltage fluctuations they see for complex stimuli. This reduced excitation is conceptually similar to less rectifying input. How does that finding relate to the current work? The lack of comparison between the model proposed by the authors and the models in the literature is surprising. Additionally, the authors draw conclusions from just one instance of voltage responses. It is conceivable that the models they explore would perform better over different regions of the parameter space. For instance, to compare between models, the authors can measure DS for stimuli used in their experimental tuning map (figure 5c) or perhaps naturalistic movies.

5. Figure 1: what is the point of arrows on panel 1g near pd and nd? are they supposed to resemble the pd direction in e? I found them extremely confusing because they point in the direction of the voltage axes. Speaking of orientation, why couldn't panel e be facing in the same orientation as panels a-c?

6. Figure 2. Can the authors show actual calcium traces? Based on their examples shown on Figure 3b, contrast responses in Tm9 ARE NOT LINEAR as they have pronounced activation for OFF stimuli but hardly any responses to ON stimuli. Given that this is the main point of the study, it is absolutely essential to show responses to individual bars. The authors never show a full response vs. contrast curve. To claim that Tm9 (and Tm2) are linear, they must show example responses at different contrast levels and provide a statistical measurement of linearity, which is not easy. How do you measure linearity with dF/F ? Is a stimulus that increases the calcium signal by 20% equal to a stimulus that decreases the calcium signal by 20%? What about a signal that increases the

dF/F by 200%? Does it have an opposing intensity signal? Going back to point 1, is linearity even important?

7. Figure 2: I found the color scheme to be unnecessarily confusing. The same color palettes are used for different stimuli (c, f) and RF properties (d, e). Also, why don't the authors show RF amplitude for ternary noise (figure 2f)?

8. Figure 3: it would be nice to see actual examples of responses to bars.

9. Figure 4:

9.1. Why don't we get to see the data for Tm1 and Tm4 ON RF properties? It would add to the difference between ON responses across cells.

9.2. Why is the number of ROIs different between ON and OFF RFs? Figure label has a somewhat cryptic indication that 'Only neurons with responses in both orientations fitted by a single Gaussian with $r^2 > 0.2$ were included'. What does it mean? Did the authors exclude all ROIs that did not produce a good RF with ON bars? If that is the case, the data is extremely misleading! To give a hypothetical example: let's assume that a cell type always produces a good response (1 dF/F) to dark bars, but only 1% of neurons in that class respond to the ON bars (i.e., 99% of the cells have 0% change in dF/F and 1% reduce their responses by 1 dF/F). The natural conclusion from this example is that these cells prefer OFF signals, as their responses to OFF are 100 times larger than their responses to ON stimulation. However, if we don't count all the cells that had 0% change in dF/F (because the cell did not pass the 'Gaussian with $r^2 > 0.2$ ' inclusion criteria, we will reach the absurd conclusion that ON and OFF responses are represented equally by this population.

10. Figure 5: I don't understand what is the purpose of this data. We know that Tm9 innervate T5 cells. Of course, they can influence T5 responses. Even the motivation (lines 191-195) makes it abundantly clear that Tm9 and T5 are linked. Would the manuscript suffer without this dataset? The only possible merit I see in such experiments is in the comparison between the effects of different presynaptic cells on T5 responses – but these experiments were not done.

11. Figure 6:

11.1. Panels a,b, it would help to add the zero line for dF/F traces.

11.2. How can you get to dF/F < -1 (panel c). Where is the corresponding data in b?

11.3. Can the reduction of RF size after CDM be an artifact of smaller RF amplitude (the iceberg effect)?

11.4. Why are some cells responding with a negative dF/F to dark stimuli in f? Are they Tm9 cells? Were not the authors surprised to see this outcome? Why isn't it discussed?

12. Line 470: I am confused by this sentence. "response median" I assume it means 'the median value between different trials, is that correct? When was the median used, and when the maximum absolute response? Would not taking the max absolute response introduce noise or bias the measurement? For example, if the responses' distribution is normal with mean=0, taking the absolute maximum would very likely result in a positive number for the cell response and not zero.

13. Almost everything in the manuscript is focused on the properties of the Tm9 cells. We get some glimpses on other excitatory cells in the system, but the contribution of their responses to T5 function is not measured (see point 10). Most importantly, the role of the main inhibitory cell CT1 in the computation of DS is not examined. The authors allude that CT1 is driven primarily by Tm9 and the reader is allowed to assume that CT1 responses are similar to Tm9. In fact, CT1 are cells with complex, long, narrow dendrites that are likely to innervate and receive inputs from numerous other neurons (Shinomiya et al 2019). How can the authors frame their study as 'we... identified the ON inputs that provide the cellular basis to implement a linear mechanism for direction selectivity in the fly OFF motion pathway.' (lines 244-5) when they clearly did not show any information about inhibition? Does CT1 provide an ON inhibitory input? Or maybe, it is a pure OFF cell? The latter is a definite possibility given that T5 is also postsynaptic to Tm9 - yet it is an OFF rectifying neuron. We know that inhibition is especially important in T4 and T5 DS computation, and thus this information is needed to support the conclusions of the study.

Point-by-point reply

REVIEWER COMMENTS

Reviewer #1 (Remarks to the Author):

Ramos-Traslasheros & Silies present results showing how the response properties Tm9 neurons influence responses of T5 direction-selective neurons. They measure calcium signals in Tm9 (and other input neurons to T5) in response to a variety of stimuli, show a correlation between Tm9 responses and downstream T5 responses, and show how Tm9 responses are modulated by the octopamine agonist CDM. They relate the response properties of Tm9 to the direction-selective computations in T5 by a numerical model.

This work demonstrates that both Tm9 and Tm2 respond to contrast increments (ON stimuli) by hyperpolarizing, in contradiction to previous studies that modeled them as only epolarizing to OFF stimuli (Serbe et al.). Tm9 neurons have receptive fields of different sizes in response to ON vs. OFF stimuli, which is an interesting result and further resolves an apparent discrepancy in the field (Fisher et al., 2015 vs. Arenz et al., 2016). It is important because it also serves as an excellent example (and warning) for how stimulus protocols can effect measured properties. The data in this paper is high quality, but there are significant issues that should be addressed in its presentation.

We thank all the reviewers for the thoughtful and detailed comments on our paper. Below, we answer all questions and comments, with our answers shown in blue.

Major

1) There were two issues with terminology that confused this reviewer.

a. First, “linear computation of direction selectivity”, referenced throughout, seems like a contradiction in terms. It is widely recognized that direction-selectivity requires a nonlinearity. This is said explicitly both in Adelson & Bergen’s seminal paper on the motion energy model and in Egelhaaf & Borst’s 1989 review. What is described here as ‘linear direction selectivity’ is the linear processing upstream of a nonlinearity that generates direction-selective signals. This needs to be made clear in the paper, starting in the title, abstract, and introduction.

b. As a related point, from the first sentence of the abstract and the introduction, this manuscript takes as canon that there is a linear transformation from stimulus to voltage in Dmel DS neurons. This should be tempered when introduced: though Wienecke argued for this, the two Gruntman papers spent substantial time on the -nonlinear- null-direction suppression aspects of their voltage measurements, caused by current shunting. Similarly, Badwan suggested that the nonlinear interactions between conductances might be responsible for phenomena that were not possible with a purely linear mechanism.

>> We thank the reviewer for this insightful criticism. We agree that we had focused too much on linear mechanisms as described by Wienecke et al., and too little on nonlinear mechanisms as highlighted by Gruntman et al. and others. In fact, our conductance model included the same nonlinear aspects in the T5 portion as previous models from the Clark and Reiser labs, and as also pointed out by this reviewer in a comment below.

Interestingly, both the linear mechanism as proposed by Wienecke et al. as well as the nonlinear conductance model benefit from negative ON responses in the inputs, arguing that our results are valid for different algorithms of DS computation. We therefore changed our writing throughout the text, from title to discussion, and adjusted for this, and also highlighted the general benefit of the negative ON responses identified here. We more extensively describe the nonlinear aspects of motion computation as described by Gruntman et al. and Badwan et al. in the introduction, and not just in the discussion as done previously. To just name a few examples:

- The title now reads “The physiological basis for the computation of direction selectivity in the *Drosophila* OFF pathway”

- We are more explicit and detailed in the introduction:
For example “*However, recent evidence based on voltage recordings argues that direction selectivity emerges from linear summing of T4/T5 inputs⁷...*”
now reads “*However, recent evidence based on electrophysiology and voltage imaging argues that direction selectivity can also emerge from linear summing of T4/T5 inputs,⁷ or nonlinearly through sublinear integration in a portion of the T4/T5 receptive fields resulting in null-direction suppression (Gruntman et al. 2018, 2019). Specifically, sublinear voltage models are supported by experiments showing that direction-selective voltage signals in T4/T5 can be generated by summing synaptic input currents arising from individual bar responses of one contrast polarity.^{8,9} In turn, a linear model predicted T5 voltage responses to moving sine wave gratings (Wienecke et al. 2018). Furthermore, a dynamic nonlinearity in T4/T5 could adjust between these integration regimes (Badwan et al. 2019). In both models, the resulting voltage is then nonlinearly transformed into a calcium signal.¹⁵”*
- We removed mentions of linear summation throughout the text, unless they were references to Wienecke et al. 2018.

c. The second terminological issue that confused me was the use of ON and OFF. In the original usage of these terms (Kuffler 1952), units were classified as ON or OFF by whether they fired to contrast increments or decrements. ON and OFF neurons later came to be neurons that depolarized to increments or decrements; the definition is independent of rectification, as for example applied to quite linear bipolar cells or to tonically firing ganglion cells (Enroth-Cugel & Robson, 1966; Werblin & Dowling 1969). It was very disconcerting for the hyperpolarization of an OFF cell (Tm9) to be called an ‘ON response’ (lines 109, 111, 135, 136, and many other places). As used in the manuscript, a perfectly linear cell would have both an ON response and an OFF response; this leads to confusion. More generally, there are ON/OFF stimuli, ON/OFF edges, ON/OFF cells, and depolarizing and hyperpolarizing responses in cells to these stimuli. These should be very explicitly defined and referred to so as not to lead to confusion based on prior definitions in the field.

>> We appreciate the suggestion and clarified terminology accordingly throughout the manuscript.

First, we are explicit about how Tm9 is responding to ON or OFF stimuli. For example, in the abstract, we changed “Tm9 ON responses” to “calcium decrements in Tm9 in response to ON stimuli”. We also avoid ambiguous terms such as “ON and OFF signals” and are now more careful with the terms selectivity/rectification. For example, “Whereas T4 incorporates ON and OFF signals...” now reads “Whereas T4 incorporates inputs from both, neurons that respond to ON and from neurons that respond to OFF signals, all major T5 inputs have been reported to be OFF-rectified”

Second, we added definitions

- “contrast increments (ON) and decrements (OFF)”
- ON bars (bright bars on a dark background) and OFF bars (dark bars on a bright background)
- ON inputs, i.e. neurons that carry information about contrast increments,
- Negative Tm9 responses to ON bars (ON receptive field) were generally wider than positive Tm9 responses to OFF bars (OFF receptive field)

2) The authors say several times, including in a discussion section, that their data imply a push-pull mechanism. I do not think the data here support this claim. Push-pull amplifiers use two transistors, one to pull voltage up, one to pull it down. In neural circuits, push-pull is canonically used to describe neurons that are excited by inputs of one polarity and inhibited by inputs of the opposite polarity – that is, the excitatory and inhibitory inputs are critical to this description, not merely depolarization or hyperpolarization. What is described here is instead more like “push-push less”, since the idea is that Tm9 is excitatory with some tonic activity that can increase or decrease with contrast decrements or increments, respectively. The reduction in excitation from Tm9 onto T5 due on an ON stimulus causes T5 to hyperpolarize relative to baseline. That model for neural interactions would not be a push-pull model. For instance, in vertebrate retina, photoreceptors release glutamate tonically, and the release can be modulated up or down; the synapse from PRs onto BPCs is not considered to be push-pull. Moreover, if push-pull applied whenever a neuron depolarizes to one polarity of stimulus and hyperpolarizes to another, then every linear visual neuron would be described as ‘push-pull’.

>> Given the lack of direct evidence about the nature of the hyperpolarizations or depolarizations observed in our data, we corrected the mentions of “push-pull” to “contrast-opponent” when addressing T5 neurons. This can be seen starting from the abstract and throughout the text.

For example:

- “direction-selective cells in the fly show push-pull properties...” now reads “direction-selective cells in the fly show contrast-opponent properties”.
- “Because all subunits respond positively to OFF and negatively to ON, the push-pull property is now satisfied.” now reads “Because all subunits respond positively to OFF and negatively to ON, T5 receives contrast-opponent inputs across its receptive field.”

3) As presented, there appear to be two important conceptual issues with the model.

a. First, the authors say multiple times that their model integrates neural inputs linearly. But their numerical Neuron model should not be linear in its conductance inputs: (a) reversal potentials for each synapse mean postsynaptic voltage saturates with increasing conductance and (b) shunting in currents can occur between synapses, so that conductances interact. This sort of passive membrane model is linear for injected currents, but not for postsynaptic conductances, which saturate and interact nonlinearly. The model does not appear to do what the authors say it does. In fact, the model used here seems very close to the nonlinear interactions used to model data in Gruntman et al. and in Zavetone-Veth et al.

>> The reviewer pointed out a potentially misleading description of the model, as it indeed integrates input currents linearly, but it is nonlinear in the conductances, like the steady state voltage solutions used in Gruntman et al. and Zavathone-Veth et al. Therefore, we have corrected the text and “linear model” now reads “model” where we refer to our computational model. The passive membrane model is build from the model approach in Gruntman et al. 2018. We extended this model to include neuronal inputs modeled from data in Arenz et al.. This is described in the methods, and is similar in aim to the model in Zavathone-Veth et al., but our model does not include a voltage-to-calcium output nonlinearity in T5 to compare model outputs to voltage recordings from Wienecke et al.

b. Second, the description of the model in the methods says that negative conductances are possible when there’s no rectification on the inputs to T5 (lines 504-507). Negative conductances are clearly non-physical, so should not be present in any biophysical model. In the case of modeling a nonrectified response, some other means must be found to keep the conductance non-negative and linear (perhaps within a given range of input contrasts). If offsets are added, then comparisons need to be made with similarly offset rectified inputs.

>> We thank the reviewer for the insight. We have now corrected the model to include an offset conductance for all neurons, linear and rectified. This offset guarantees non-negative conductances in a contrast range of -1 to 1. In addition, we simplified the linear temporal filters of the input neurons by using a high-pass filter for Tm1, Tm2, and Tm4, and a low-pass filter for Tm9 and CT1, without changing the original parameters from Arenz et al. This is now more extensively described in the methods, and produces biologically plausible neuronal Tm responses to full field flashes (data not shown). None of these changes altered any conclusions that we drew in the manuscript.

4) The DSI in figure 1 was measured by finding the amplitude of the response component at the temporal frequency of the input grating. This is a nonlinearity applied to the voltage trace, but it’s not clear what kind of biophysical mechanism could perform this operation by isolating a specific frequency. The nonlinearity chosen here determines the direction-selective properties of the response if upstream processing is mostly or entirely linear, so it’s important to choose a nonlinearity that could be plausibly (or cartoonishly) implemented in a biological system. The authors should justify the specific nonlinearity they choose.

>> The Fourier analysis used in Fig. 1 is a standard procedure in linear systems analysis, that has been applied in classical vision research (Mecher et al. (1998) JNeurosc, Movshon et al. (1978) JPhysiol) and more recently in Wienecke et al.

Other readouts typically used in fly vision include the mean of the response, or the peak response. We favor Fourier analysis for signals that oscillate about a (close to) zero baseline and still use this as read-

out in the main figure. However, we generated a new Supplementary Fig. S1 which shows the quantification of direction selectivity when using the peak amplitude, as well as the mean after rectifying and squaring the voltage signal. The latter has been used to model the calcium output (e.g. in Leong et al and Zavathone-Veth et al.). The conclusions in our manuscript stay the same when using these other read-outs.

5) The correlation between Tm9 and T5 receptive field sizes is interesting. As the authors note, each Tm9 is presynaptic to 4 T5 cells, and I agree with the argument that their oriented stimulus choice removes two of these from consideration. However, since the authors record in the T5 dendrites and each T5 dendrite spans 3 columns, aren't the recordings in a single column really from up to 6 different T5 cells (say 3 each from LP layers a and b)? I believe this is true, though it may depend on the orientation of the recording plane. S1e doesn't really address this point because it looks at correlation in RF center over many ROIs. Really, one might want to know the distribution of RF center differences, which frequently look like they're off by 5 degrees. I'm not sure one even expects connected Tm9 and T5 cells to have peak OFF bar responses at the same position, does one? If the authors agree that connecting Tm9 to T5 responses is nontrivial here, then this section could use substantially more explanation and caveats. It would also help to know more about the variability here: was it within fly or between fly? Was it correlated or uncorrelated with position in lobula, or with position on the screen?

>> As suggested, we looked more carefully at the distribution of RF centers. As now shown in Supplementary Fig. 5, receptive field centers are not often off by 5 degrees, but the large majority lies within 2 degrees of each other. Therefore, Tm9 and T5 receptive fields are indeed highly overlapping, and argues for the validity of our analysis.

In terms of variability, our visual impression is that there is both, within fly and across fly variability. Analysis also shows that in some flies, receptive field widths are wide spread, and they are narrow in others (Figure R1). The experiments done here are difficult experiments, with low signal to noise ratios, especially for jRGECO1a. This means that we only managed to acquire a few or even single Tm9-T5 pairs per fly, and this does not allow us to look at correlations within one fly in a statistically meaningful way. To address variability across flies, we analyzed correlations after averaging cells within one fly. Correlations remained positive and are significant for the horizontal FWHM comparison, suggesting that there is variability across flies, in addition to our already reported variability across ROIs. This is again with a sample size (n=8 flies) too low to test for correlations in a really meaningful way.

In terms of position within lobula plate or screen, we recorded from the same position within the lobula as much as possible in this study, and thus do not know how the observed variability holds across the visual system of the fly. We did not find any consistent correlation with position on the screen. We also think that especially the question about variability within the fly visual system is a really interesting one, and it is intriguing that Tm9 variability in terms of its synaptic connectivity across 7 neighboring columns was reported by (Takemura et al. 2015 PNAS), and that recent single cell RNAseq studies by the Desplan and Zipursky labs identified two dorsal and ventral Tm9 subtypes. We mention this in the discussion and are planning to address this further in follow-up studies, e.g. by reconstructing Tm9 and its inputs at different positions across the medulla within EM datasets.

Figure R1. Correlations across flies and within flies of receptive field size (FWHM) from T5 and Tm9 ROIs shown in Fig. 6f. (a, b) Scatter plots of receptive field sizes of T5 vs Tm9 measured as FWHM from Gaussian fits to responses to horizontal (a) and vertical (b) OFF bars. Dots indicate fly means, and horizontal and vertical bars indicate the range of values per fly across the respective dimension. Pearson's correlations of fly means are shown on top, with the corresponding 95%-confidence interval computed via bootstrap. **(c, d)** Correlations within flies, for two example flies, one with a wider spread (c) and one with a narrower distribution. ROIs from the example fly (red circles) are plotted on top of ROIs from all flies (gray circles). (d). Pearson's correlations and p-values given by t-test (MATLAB's corrcoef command).

6) The last section title of the results says that behavioral state modulates Tm9 response structure. To make this claim, the authors have added a global octopamine receptor agonist while imaging. Thus, the behavioral state claim is too strong and should be softened, since the authors could have run alternative experiments to make this claim directly (as in Strother et al. or Chiappe et al.). (As an aside: Is this CDM result consistent with Arenz's results? I recall they also seemed to find deeper inhibitory lobes with CDM present. The authors might remark on similarities.)

>> We softened the section title of the results as suggested by the reviewer, which now reads "Octopamine signaling sharpens the ON receptive field of Tm9". We additionally made the concluding statement of this paragraph more precise: "Taken together, our data show that octopamine signaling, mimicking an active behavioral state, sharpens the spatial receptive field of ..."

The reviewer is correct that, upon visual inspection, deeper inhibitory lobes appear to be present in Tm9 upon application of CDM (and lower inhibitory lobes in other OFF pathway neurons) in Fig. 4 of Arenz et al. 2017. However, Arenz et al. write that "In the OFF pathway, the results were very similar. The spatial receptive fields appeared unchanged by CDM for any of the columnar input neurons (Figure 4D)". Since this is not statistically tested, and the focus of the Arenz study is on temporal properties, we cite this work extensively when discussing temporal properties, but not for a more extensive comparison of the results on the spatial receptive field structure.

Minor

1) 'Linear' is used in many instances, and it was not always clear what the linear is in reference to. For example, the authors discuss linear summation of inputs in T5, linear responses of T5 to stimuli, linear transmission from Tm9 to T5, and linearity of responses in Tm9. There are several places, especially related to T5, where I found it ambiguous whether the authors meant linear in space or linear responses to localized stimuli or linearity of inputs to T5.

>> We are more conservative with the use of the word "linear". We for example only use it for models when referring to Wienecke et al., and in relation to the contrast dependency of the model neurons in Fig. 1. We no longer call Tm9 responses linear but describe that Tm9 shows increases in calcium signals to OFF and decreases in calcium signal to ON, and are more explicit throughout the text.

2) Statistics: I believe the authors treat individual ROIs as statistically independent. It seems as though there are likely to be correlations within flies, though, so is this warranted? The low p-values in Figure 6c, for instance, could result from treating correlated data as independent.

>> To address this comment, we now used statistics that consider both within-fly and across-fly variability. In line with the visual impression, both nested permutation testing and a nested ANOVA show that there is a significant effect of CDM application, but also between-fly differences. Both tests produce the same result, namely that all effects that we previously described to be significantly different still are so, with exception of the OFF receptive field surround amplitude, and width of the OFF receptive field center. We now use nested permutation testing for analysis within the manuscript, adjusted the text accordingly, and added a detailed description of the statistics to the manuscript. We did not add across fly statistics to the correlations shown in Fig. 6E (now Fig. 7e), but the line across ROIs indicates a global trend.

3) Lines 66-70: Authors state here that ON inputs to T5 are required. Cells that transmit information about contrast increments are required, given Wienecke's result, but since OFF cells need not be rectified, they could suffice. This and several other comments below are examples of the terminology issue as noted above.

>> We addressed this throughout the paper, see our response to major point (1). This specific sentence now reads: “However, information about contrast increments (ON inputs) are required ...”. This also defines our subsequent use of the term “ON input”, which we deem useful to avoid saying “information about contrast increments” or “information about ON stimuli” each time.

4) Line 71: ON inputs to T5. Responses to ON stimuli? ON cells? It’s a bit ambiguous here, and elsewhere, and it would be profitable to eliminate this sort of ambiguity.

>> see previous comment . This still reads “ON inputs”, but is now defined just above.

5) Line 74 and elsewhere: “OFF direction-selectivity” – not entirely clear how this use of ON and OFF differs from others. T5 cells respond to moving dark edges, but as the authors point out, their linear receptive fields have positive and negative regions.

>> We now avoid the term “OFF direction-selectivity” throughout the paper. This specific line now reads “direction selectivity in the OFF pathway”. Overall, when we talk about OFF direction-selective cells, we always mention T5. We also improved the definitions of ON and OFF responses and ON and OFF pathways in the second paragraph of the introduction.

6) Line 102: “OFF-rectified inputs”? When ‘rectified’ is not specified, does that mean it’s a potentially linear OFF input?

>> As the reviewer also pointed out in major point (1), the definition of ON or OFF is independent of rectification, and we now use it this way throughout our manuscript. We now only use the term “OFF-rectified” if we specifically mean neurons that only depolarize to OFF, and are not linear, and replaced the incorrect use of the term “OFF-selective” with “OFF-rectified” in various places, including the abstract.

7) Line 117: ON inputs provided by linearizing an OFF excitatory input neuron? This is a confusing description. Line 119, too.

>> We removed the term ON input from these two instances. Previous line 117 now says “This suggests that ON information from additional T5 inputs is needed ...” and the section title from previous line 119 now reads “Tm9 provides ON information required by T5”.

8) Lines 124, 172, 253: “ON input required” – figure 1 shows that linearity in Tm9 is sufficient to explain some measurements in T5, but not that it is required.

>> We toned down all of these statements, and one further sentence, as follows

- required by linear models of motion computation > motivated by linear models of motion computation (previous line 124)
- required by a linear mechanism of direction selectivity > implementing a linear mechanism of direction selectivity. (previous line 170)
- ON inputs at the central and leading sites are required for the proposed linear mechanism of direction selectivity > ON inputs at the central and leading sites can explain the proposed linear mechanism of direction selectivity (previous line 172)
- we sketched a minimal model that exposed the need of > we sketched a minimal model that exposed the benefit of (previous line 253)

9) Line 125: reference to non-existent figure 7.

>> This was meant to be reference #7 (Wienecke et al.). Now fixed.

10) Line 136: Tm9 ON responses. Calling these “ON responses” is pretty confusing.

>> We now describe this in more detail: “Negative Tm9 responses to ON bars (ON receptive field) were generally wider than positive Tm9 responses to OFF bars (OFF receptive field), both for horizontally and vertically oriented bars.”

11) Question: something similar to this phenomenon was proposed for T4, which can surprisingly be depolarized by presenting dark stimuli offset from its RF center (Agrochao et al.). That paper suggested this might be disinhibition via Mi4, which is not entirely rectified and represents a possible pathway for

OFF information to reach T4, beyond Mi9. Could the authors comment on whether these are related properties in the two pathways?

>> The reviewer raises an interesting parallel. According to our new data (Fig. 5), CT1 neurons respond positively to dark bars, but have a small positive response to bright bars away from its OFF receptive field center, and zero or slightly negative responses to bright bars in the location of its OFF receptive field center. Thus, while the ON response profile of CT1 is not a clear Gaussian-like receptive field like the ones of Tm9 and Tm2, CT1 can contribute both OFF inhibition and ON disinhibition at the preferred site of T5 receptive field, similarly to Mi4 for T4.

We added the following sentences to our discussion:

“Interestingly, a recent model of illusory motion perception in *Drosophila* required that the inhibitory input at the leading site of T4 (Mi4) responded linearly to convey inhibition to ON and disinhibition to OFF (Agrochao et al. 2020). Analogously, the inhibitory input on the leading site of T5, CT1, provides inhibition to OFF and disinhibition to ON (Fig. 5)(Takemura et al. 2017). This suggests T4 and T5 at least share a similar input mechanism on their trailing sites.”

12) Line 152: The resolution of the RF size discrepancy in the literature is satisfying and nicely shown.

>> Thank you.

13) Figure 2d: could emphasize these are both spatial dimensions.

>> We added ‘Position x’ and ‘Position y’ labels to the schematic to illustrate this, and also added “spatial” receptive field to the figure legend

14) Figure 2d, 3rd column: the dark bar and flicker receptive field peaks appear offset in x. Is this remarkable?

>> These experiments, showing various stimuli to the same fly, require very long recordings (~1.5 hours per fly), and we cannot fully exclude that there is some movement of the fly. We did not observe that the offset seen in this specific example was a general trend across our data set.

15) Figures 3e and 2cf show same information, but different plots are used. Might make it easier on readers to use same presentation?

>> We changed plot 3e to match plots shown in Fig. 2c, f.

16) Line 169 (and elsewhere): why is this receptive field called “inhibitory”? The hyperpolarization of T5 to ON stimuli doesn’t require an inhibitory ON input; as the authors show, it appears like to be an excitatory OFF cell responding at below baseline to ON stimuli

>> The reviewer is correct, we meant to say that the response is negatively correlated with the stimulus, and now call this an “negative ON receptive field” instead of an “inhibitory ON receptive field” in previous line 169 and other parts of the manuscript. We also added an explicit description of the receptive fields when we first describe them in Fig. 2: “Negative Tm9 responses to ON bars (ON receptive field) were generally wider than positive Tm9 responses to OFF bars (OFF receptive field)”

17) Fig 5d: p-value is cut off for me.

>> Here, out of $1e6$ random shuffles, the mean of none of the shuffles was ever higher than the real mean value. So the p-value is indeed zero (# shuffles with mean smaller than the original mean / # shuffles). To clarify this, we added this information to the figure legend.

18) Fig 5c: why take the amplitude of the component at that temporal frequency? This is sort of the opposite of what Strother 2013 did for analyzing similar data – that paper evaluated the mean response over time to judge bandpassing properties.

>> We were inspired by different classical papers analyzing the responses to periodic stimulation in visual cortex, including for example Mecher et al. (1998) *JNeurosc*, Movshon et al. (1978) *JPhysiol*, Sriram et al. (2016) *JNeurophysiol*, also used in some *Drosophila* vision literature, e.g. Wienecke et al. (2018) *Neuron*.

19) Fig 6bd: why can't we see the significant sharpening of the ON-bar receptive field width in the mean traces in b? I'm surprised.

>> While the change in the ON receptive field size might not be prominently noticeable in Fig. 6b, d, (now Fig. 7b, d) it is there, and it is significant. What might help here is to mention that fitting a difference-of-Gaussians quantifies the shapes of the curves by amplitude and width for each ROI, thus a change in width is better visualized by normalizing each ROI tuning curve by its response range (max response - min response). Moreover Fig. 6b (now Fig. 7b) compared the means of the raw traces, while Fig. 6d (now Fig. 7d) compared the mean after fitting. Therefore, we included a figure for the reviewer (Fig. R2) displaying the receptive field changes by first normalizing each ROI receptive field to its range and then averaging across orientations, to finally average across ROIs. We must remark, the receptive fields throughout the paper are only shown aligned for visualization purposes. We are happy to include this analysis if the reviewer feels strongly about this, but thought that it is sufficient to show the quantification in Fig. 6d (now Fig 7d), and Supplementary Fig. 6b.

Please note that, in response to a comment by reviewer 3, Fig. 6a-e (now Fig. 7a-e) was updated to constrain the amplitude ranges of both center and surround to values close to the peak response. The conclusions regarding the sharpening of receptive field centers for ON and OFF bars, and the reduction in receptive field center amplitude for OFF bars remain unchanged.

Figure R2. Visualization of normalized receptive field traces from current Fig. 7a,b. (Top, bottom) The receptive fields of each ROI in Fig. 7a,b were normalized to their response range per ROI (max response - minimum response) to visually match more closely the difference-of-Gaussian analysis in Fig. 7c,d. Data were averaged across orientations, and across ROIs to generate mean traces. (Middle) The mean traces are overlapped for better visualization of the receptive field size changes. Control (blue, n=46), CDM (orange, n=49).

20) Fig 6e caption: “inset shows augmented version of dotted box”? What does this refer to? I can't figure it out.

>> The figure panel contains an inset showing the data with low dF/F values (-0.7 to -0.2 on both abscissa and ordinate). We added lines to link the inset to the data shown in the main figure panel (now Fig. 7e).

21) Line 256, line 292 as well: Tm9 and 2 have inhibitory ON receptive field components – I don't think this paper provides evidence that these are inhibitory components, just that the cells hyperpolarize in response to ON stimuli.

>> this now reads “negative ON receptive field components” in previous line 256, and “Negative ON responses...” in previous line 292

22) Line 288: “close to complete” – it seems premature to declare this.

>> we weakened this statement to say “is now well understood”

23) Line 292: These are not proven to be inhibitory. They’re hyperpolarizing.

>> Same as 21). We changed this to “Negative ON responses in OFF pathway neurons”. (because we are measuring calcium, we are avoiding the term hyperpolarizing)

24) Line 293: Avoid editorializing on the importance of the work.

>> We deleted this statement. The sentence now reads “Our work characterizes the response properties of the main input neurons to direction-selective T5 cells.”

25) Line 296: “ON input to OFF direction-selective cells”. I think this framing is confusing, given the use of this terminology in the field.

>> this now reads “ON input to direction-selective T5 neurons of the OFF pathway”

26) Line 354: swap ‘primate’ for ‘human’?

>> done

27) The authors write that the variability in Tm9 RF size shows that it must be important for motion detection, but I think this is not necessarily true. Tm9 could feed into several downstream circuits, and could have properties related to non-DS functions. So to make this claim, one would need to demonstrate why this improves motion detection; its mere existence doesn’t demonstrate that it is a circuit feature that improves motion detection.

>> This is correct and was also pointed out in the first major comment of Reviewer #2. We answered this in detail below, and changed the statement about the variability being important for motion detection to “the variability observed in receptive fields is passed on to downstream computation”.

28) Line 523: Here it says that the DSI is the sum of vectors divided by the sum of amplitudes; to get a scalar out, it must be the amplitude of the sum of the vectors, divided by the sum of the amplitudes.

>> We corrected this.

29) In several figures (Fig. 2b, 4b-i), the number of ROIs for the light bar RF measurements are substantially smaller than for the dark bar RF measurements. But not in 6ab. This seems to be because only ROIs with significant responses are shown in Figs 2 and 4. Does this artificially increase the mean responses to ON stimuli? For instance, if one includes all ROIs that respond to either ON or OFF stimuli, is the ON-dependent hyperpolarization smaller, since one averages in many more responses of 0? Is this an important issue?

>> The reason why much fewer ROIs were shown in the main figure is because we can only quantify parameters from a meaningful fit to the data, which we defined as a threshold of $r^2 \geq 0.2$ from fitting a Gaussian to each tuning curve for each orientation. Furthermore, a response quality index was used as a measure of trial-to-trial variability to exclude noisy cells regardless of the fit (after Baden et al. 2016 Nature). This is now both explained in detail in the methods.

To test if this artificially increases mean responses, we also analyzed data of Fig. 4, which includes all four Tm neurons, without applying any quality control. As now shown in new Supplementary Fig. 4, negative responses to ON are still prominent in Tm2 and Tm9 neurons. Even when averaging all responses ($n=98$ for Tm1, $n=152$ for Tm2, $n=55$ for Tm4, $n=108$ for Tm9), the mean amplitude of the negative ON response is similar to the ON response shown in the main figure. This is true because the ON response is visible across all individual ROIs, including noisy ones. In addition, this analysis shows that there is no negative calcium signal in response to ON bars in Tm4, but Tm1 shows a negative peak in the tuning curve, mostly visible for vertical bars. However, because we shift traces (purely for visualization) after fitting a Gaussian for all ROIs, we cannot exclude that this is an artifact of alignment to the peak.

To analyze the effect of CDM application on the receptive fields, we only included cells for which ON and OFF responses passed response quality control criteria, thus resulting in a rather conservative analysis.

Moreover, here we fit a Difference-of-Gaussians to the receptive fields, which might produce better fits to the data than the single Gaussian used in previous figures, but have now also included the results of using a single Gaussian (Supplementary Fig. 6). All of this is now described in detail in the methods.

Reviewer #2 (Remarks to the Author):

In their manuscript entitled „The physiological basis for linear computation of direction selectivity”, Giordano Ramos-Traslosheros and Marion Silies use in vivo two photon calcium imaging to explain how linear summation in T5 cells leads to direction selectivity in the *Drosophila* OFF pathway during motion vision. By revealing unexpected ON-responses for both Tm2 and Tm9 neurons, the authors reveal that linear summation in the OFF pathway might indeed be more similar to the ON pathway (culminating on T4 cells), than previously known. Despite superficially sounding like ‘filling in the gaps’ his finding is very relevant in several ways. First, it shows that T4 and T5 inputs are most likely not using fundamentally different mechanisms, since they both combine ON and OFF inputs along the dendritic tree, resulting in a push-pull configuration. This was previously not known. Hence, this work provides important progress towards working out fundamental principles underlying motion vision. This is further highlighted by the multiple parallels to direction selectivity of mammalian cells, which are being worked out by the authors throughout the paper in an exemplary fashion. The data presented are of very high quality and to the most part very clear (see my comments below, for improvements). Importantly, the authors use GCamp6f as an indicator and successfully visualize hyperpolarization, thereby validating previous experiments from other groups that used either voltage imaging or electrophysiology. Besides providing such synergistic validation, the data provided here will be of great interest to anyone performing similar experiments/answering similar questions in a different experimental context. Furthermore, the authors use two color imaging in order to confirm the unusual properties of Tm9 (by directly comparing it to Tm4), and experiment that I would have requested, although knowing that it would be technically very challenging. It speaks for the high quality of the data presented here, that such an experiment is being provided.

I have quite a few comments regarding the Figures and how they could be improved, as well as a few comments on the text. If these points were to be addressed, then I recommend this manuscript for publication in Nature Communications.

Major points:

- The authors use the variability of Tm9 responses as a clever feature to demonstrate information flow from Tm9 to T5. In the process, some statements are exaggerated, in my opinion. For instance in lines 192/193, the authors speak of ‘biological relevance’. I find this term misplaced here. The upcoming experiment does not technically test for this, but merely investigates whether this feature is transmitted to the post-synaptic T5 cell. The same kind of sloppiness arises again in line 211, where the same variability is identified as being ‘important’ for downstream computations. Again, this is an over-interpretation of the data, since all it shows is that the variability is maintained. This might very well be important, especially since it makes sense computationally, but it hardly provides proof supporting such a strong statement. Finally, in line 259, the authors identify the tight coupling between Tm9 and T5 as being ‘critical’ for motion computation. Again, this is over-stated. The data is in line with the model, the response properties of the cells correlate, but is no proof for Tm9 being critical. These sentences should all be softened.

>> The reviewer is correct, and we softened the specific statements:

- previous line 192/193: “We asked whether this is passed on to downstream computations”
- previous line 211: “the variability observed in receptive fields is passed on to downstream computation”
- previous line 259: “Tm9 exhibits variability in its physiological receptive field properties, which are tightly coupled with downstream T5 properties.”

- Line 304: The authors state that T5 do not manifest Calcium responses to ON bars. Is that really true? When I clinch my eyes, I seem to see responses in Figure S1. Is there not even a trend? Can the authors comment on that? Regardless, I recommend moving the data from supplemental Figure S1 into Figure 5 (see below), since it is being discussed in detail, so it is weird not to be shown in the main Figures.

>> We have been debating the same question. To visualize receptive fields in these plots, we shift curves of individual ROIs based on a Gaussian fit to the each ROI's response. This could lead to this "trend" towards a negative mean. When quantifying this, we found that these responses are overall tiny (~ 2% dF/F) and only in less than half of the cases, the peak amplitude of the fit is significantly higher than the mean tuning curve (see Figure R3).

Because this is not really relevant to the story, we decided to not expand on this in the manuscript, but changed the description of these panels to "Both Tm9 and T5 responded positively to OFF bars, whereas only Tm9 showed prominent negative responses to ON bars". This still demonstrates that we are able to separate Tm9 and T5 responses spectrally.

Figure R3. Quantification of T5 responses to ON bars shown in Fig. 6c. (a) Distribution of amplitudes of the negative Gaussian fits to each T5 tuning curve from responses to horizontal and vertical ON bars. $n = 41$ (b) P-value distribution from Wilcoxon's signed-rank one-tailed paired. This tested whether the negative amplitude from the fit was significantly smaller (larger negative peak) than the mean of the tuning curve it was derived from. (c) Proportion of significantly different peak responses as determined by a Gaussian fit, potentially indicating negative T5 ON responses ($p < 0.05$, white) and non-significant ROIs ($p \geq 0.05$, gray).

- Colors used in the Figures: I find it very confusing when purple and green are being used within the same panel, while encoding two different things, i.e. showing traces of two different cell types like Tm9 and T5, while at the same time encoding depolarization versus hyperpolarization for both cell types. This needs to be changed. I recommend using light blue and light orange for the traces, while maintaining the color code for activation and inhibition, or vice versa. Maybe changing the color code of the latter makes even more sense, since purple and green are also used when labeling the cell types using immune histochemistry.

>> As suggested by the reviewer, we maintained color code for positive and negative calcium responses, and changed the color code indicating cell types. We are now keeping colors for cell types consistent across all figures, with Tm9 being light blue, and T5 being gray.

- As stated above, the data from supplemental Figure S1 should be incorporated into Figure 5, since it is quite relevant and is being discussed rather prominently in the context of Figure 5.

>> We moved previous Figure S1a-d to the main figure.

- Figure 6F: Can the authors comment on the inverted response traces of quite a few individual Tm9 cells when stimulated with OFF and ON stimuli? Maybe I am not on top of what is known about Tm9,

but some of these cells seem to hyperpolarize in response to OFF and depolarize in response to ON. How come?

>> We apologize that we failed to point this, but these are likely cells that are not directly looking at the screen, and the responses of opposite sign are consistent with an inhibitory surround. We previously described this inverted response type in our initial characterization of Tm9 in Fisher et al. (2015) *Curr Biol* 25: 3178-89 (Figure S3). We now mention this in the figure legend: *"Few cells show an increase in calcium signal in response to OFF. This inverted response type likely represents cells not directly looking at the screen (see also Fisher et al. 2015)."*

- I think a final, summarizing Model figure linking back to Figure 1A would be very helpful, summarizing the ON/OFF inputs along the dendrite, and maybe even the octopaminergic modulation.

>> We have added a summarizing model to Figure 7h-j.

Minor points:

- Line 16: "...is passed ON to..."

>> fixed

- Line 31: Wouldn't "...AT LEAST three synapses away from photoreceptors..." be more accurate? There is always direct and indirect pathways, no?

>> yes, changed as suggested.

- Line 36: Should electrophysiology from previous studies also be listed here, in addition to voltage imaging?

>> We meant to include both electrophysiology and voltage imaging in the term "voltage recordings" and had cited Wienecke et al. 2018, Gruntman et al. 2018, Gruntman et al. 2019. To make this more explicit, this now reads: "However, recent evidence based on electrophysiology and voltage imaging argues that direction selectivity emerges from linear summing of T4/T5 inputs,⁷⁻⁹"

- Line 166: Rather than starting the sentence with 'because' I would start it with 'since'.

>> changed as suggested.

- Line 169/170. It would be more accurate to speak of Tm9 being a very suitable candidate for providing ON input onto T5

>> yes, changed as suggested.

- Line 199: Rather than starting the sentence with 'because' I would start it with 'since'.

>> changed as suggested.

- Figure 1A: Please move the explanation of excitatory and inhibitory synapse arrows somewhere else (between the T4 and T5 dendrites?), since the way it is depicted now, their current position erroneously insinuates that they are homologous to the PD, in the OFF pathway.

>> done

- Figure 1G: Please move the previously published voltage imaging traces BELOW the outputs of the three models, simply because it is important to understand which traces belong to which model. This is most easily done by placing them right underneath each other.

>> done

- Figure 3B-D: I would find it more consistent if all Tm4 data was placed in the left column, and all the Tm9 data was placed in the right column. As it is now, cell type specific data is presented in columns (B) and rows (C,D) which is rather confusing.

>> We changed the order of the panels as suggested by the reviewer.

Reviewer #3 (Remarks to the Author):

In this work, Ramos-Traslosheros and Silies study the mechanisms of direction selectivity (DS) in the T5 neurons in the fly. First, they construct a model of T5 dendrite receiving its known presynaptic inputs (Tm9 near the tip, Tm1, Tm2, Tm4 in the middle and inhibitory CT1 at the base of the dendrite). They stimulate the model with drifting gratings and observe the resulting DS. They show that the best DS is achieved with linear (non-rectifying) Tm9, Tm2 and CT1 cells vs the alternatives of all rectifying presynaptic cells of just rectifying Tm9 (figure 1). Second, to examine the level of rectification in presynaptic cells they express calcium reporters and map the receptive field (RF) of the cell with vertical and horizontal bars (figures 2-4). They find that Tm9 responds well to light bars with reduced calcium signals, irrespective of bar orientation (Figure 2e). Additionally, the spatial extent of RF size is much larger for ON bars or white noise (Figure 2f). Using a combination of red and green calcium indicators, the authors then compare the RF properties of Tm4 and Tm9 cells. Using such 'paired' recordings they confirm that Tm4 is a rectifying OFF cell. Its Rf size matches the size of the Tm9 OFF component (Figure 3). In figure 4 they combine RF analysis for Tm1, Tm2, Tm4 and Tm9 on a single panel and show that Tm2 and Tm9 are the only cells with ON RF components and these components extend further in space than the OFF RFs of all cells. Next, the authors again use double recordings to measure the spatiotemporal characteristics of the Tm9-T5 cells with similar RF locations (supp figure) and reveal a Pearson correlation of about 0.4 between the widths of the RFs and highly similar responses to gratings (Figure 5). Last, CDM application to activate octopamine receptors reduced the amplitude of the responses in Tm9 and the size of the ON center component of the RF. I find the work to be interesting, but I do not think that the experiments and the framing of the paper as presented in the current format warrant the conclusions reached by the authors. Specifically:

>> We thank the reviewer for the generally positive assessment of our work . We hope we can clarify the remaining points below.

1. First, the motivation for the study and the title. What do the authors mean by 'linear computation of DS'? As I understand it, the main point of the study is to understand the contrast responses of T5 presynaptic neurons. In the text I could see two different concepts of linearity:

(1) non rectifying contrast responses (lines 41-43)

(2) previous work (notably Gruntman et al 2018, 2019, Wienecke et al 2018) showed that the computation of DS is performed by a linear postsynaptic integration over presynaptic units (lines 35-40).

The authors seem to assume that (1) is required for (2), but why is that so? Would the results of the present study be invalidated if it turns out that Reichardt and Hassenstein were right all along? A much better motivation for this study would have been an exploration of contrast dependency of DS in T5 neurons, as this is what the model proposed by the authors (figure 1) shows. At the very least, the manuscript requires a major rewrite to address this point.

>> The motivation of our study was to understand the physiological basis behind motion computations. The requirement for (1) non-rectified contrast responses is given by the data in Wienecke et al. 2018, which show that T5 neurons fulfill both (1) and (2). While computationally, direction selectivity can be achieved without (1) as shown in Gruntman et al. 2019, a complete description of the physiological basis of T5 direction selectivity should consider (1). Furthermore, our model in figure 1 model suggests that looking at the contrast responses of presynaptic neurons is critical in this context. We changed the title to "The physiological basis for the computation of direction-selectivity in the Drosophila OFF pathway". We also edited major parts of the manuscript, as suggested by the reviewer, and hope that it now better reflects our data.

2. Line 88: The DS effect of negative ON excitation is not equal to an ON inhibitory input, as is clearly shown by Gruntman et al 2018.

>>> This was the approximation used in the cited model by Zavathone-Veth et al., which used three inputs along the preferred motion axis arranged as: OFF inhibition, ON excitation and ON inhibition to model T4 responses. As the reviewer points out, Gruntman et al. 2018 shows inhibition and not just release of excitation is required to explain their T4 data. However, the ON inhibitory input in Gruntman

et al. 2018 is restricted to the T5 leading site, whereas this sentence was specifically referring to the trailing site, getting inputs from Tm9.

Zavathone-Veth et al. approximated a T5 neuron by inverting the polarity of all (T4) inputs resulting in ON inhibition, OFF excitation and OFF inhibition. However, the ON inhibition to T5 (equivalent to OFF inhibition to T4) had not been experimentally reported, but would further be consistent with Wienecke et al. data. Finally, the Tm9 negative responses to ON could be a release of excitation to T5, which implies T5 achieves DS in an analogous way to T4, but the mechanism is not merely a polarity-inverted version of T4.

3. The model (Figure 1)

3.1. While the model examines voltage attenuation in a T5 dendrite, the model itself is just a 13.3 μm long, 0.2 μm wide cable. From a biophysical standpoint, there is no significance to the spatial location of inputs in this tiny patch of membrane. The model doesn't approximate the actual voltage propagation in a T5 dendrite, which is highly branched and has one end that is leading to release sites while the other end gets to the soma. If the authors consider the spatial arrangement to be essential and want to consider dendritic voltage attenuation (Rall's model of DS) they should at least mimic this morphology.

>> We appreciate the suggestion. The reviewer accurately infers the spatial location of inputs in the membrane is not of large significance. But, rather than dendrite morphology, it is the visual spatial location of the inputs that determines the direction selectivity, as the three inputs come from three different eye columns. In fact a single compartment model is a good description as Gruntman 2018 showed “we expected that a single -compartment neuron simulation should also be able to capture the response dynamics of T4. Indeed, this simpler model reproduces the moving bar responses of T4 with only negligible differences to the multi-compartment simulation results”. A single-compartment model of T4 is also implemented by Zavathone-Veth 2020. Thus, we consider the model approximation useful as it is.

3.2. Is there a reason to use traces from Wienecke et al 2018 which were based on voltage-sensitive dyes? Gruntman et al 2019 performed direct recordings from T5 neurons.

>> The traces in Wienecke et al 2018 include both gratings, and responses to bright and dark bars from the axon terminals of T5. The response to the ON component is critical for the data shown in Wienecke et al, and was direct motivation for our work. However, regardless of the model and traces used for visualization, the model in Figure 1 was meant to motivate our experimental work, and subsequent data confirms input neurons to T5 convey ON information.

4. Using conductance-based models, Gruntman et al 2019 demonstrated that reduced excitation could not fully replicate the voltage fluctuations they see for complex stimuli. This reduced excitation is conceptually similar to less rectifying input. How does that finding relate to the current work? The lack of comparison between the model proposed by the authors and the models in the literature is surprising. Additionally, the authors draw conclusions from just one instance of voltage responses. It is conceivable that the models they explore would perform better over different regions of the parameter space. For instance, to compare between models, the authors can measure DS for stimuli used in their experimental tuning map (figure 5c) or perhaps naturalistic movies.

>> The reviewer correctly points out that direct inhibition is critical according to Gruntman et al. 2019. However, Gruntman considered only OFF-rectified inputs: broad excitation on the trailing site, and narrow inhibition on the leading site. While we had already mentioned in the discussion (previous lines 278-282) how our results are consistent with Gruntmann, we now added direct references to Gruntman in the introduction and results. For example: “A model that received only OFF-rectified inputs produced direction-selective responses, reproducing a mechanism consistent with the one described in (Gruntmann et al. 2019), which requires OFF excitation over the trailing and central sites (here Tm9, Tm1, Tm2, and Tm4), and OFF inhibition in a smaller region on the leading site (here CT1).” Our work shows CT1 could provide the direct OFF inhibition to T5, but also shows that release of excitation from Tm9 and Tm2 during ON stimulation is an important component of T5 receptive fields. This latter finding bridges the conceptual gap between Gruntman et al 2019 and Wienecke et al. 2018.

Additionally, to avoid misleading readers to think we based comparisons on one instance of voltage traces, we included the tuning curves used to calculate the direction-selectivity index (Supplementary Fig. 1), which consist of responses to 16 different motion directions, but for simplicity still show only two representative directions in Fig. 1. We have now also included how the model quantification is robust to other parameters (see Reviewer 1, comment 4) in a new Supplementary Fig. S1.

Instead of fitting parameters, we decided to use published parameters for the physiological properties of T5 input neurons (Arenz et al. 2017), approximate dendrite and synaptic parameters (Gruntmann et al. 2018), and relative synaptic connectivity (Shinomiya et al. 2019). This way, we tried reducing both biases and assumptions in the model. While we agree that the models could perform differently under different parameters, the constraints set here are biophysically plausible and consistent with previous literature, and thus allow a fair comparison across models. We further varied the ratio of excitation to inhibition (our only free parameter), and show the results are robust to this parameter (Supplementary Fig. 1f-h). Finally, as suggested by the reviewer, we added the performance of the models across different temporal frequencies (Supplementary Fig. 1i-k). Interestingly, direction selectivity is highest when T5 receives linear inputs in the three sites, followed by the two linear input sites in the calcium responses. We thank the reviewer for the suggestions to improve the model which is used to motivate our experimental work investigating the biological basis of motion computation.

5. Figure 1: what is the point of arrows on panel 1g near pd and nd? are they supposed to resemble the pd direction in e? I found them extremely confusing because they point in the direction of the voltage axes. Speaking of orientation, why couldn't panel e be facing in the same orientation as panels a-c?
>> The arrows were meant to illustrate that the PD was downward motion, and ND was upward motion. We apologize for the confusion and added a little stimulus schematics that shows this. We also turned panel e, as suggested by the reviewer.

6. Figure 2. Can the authors show actual calcium traces? Based on their examples shown on Figure 3b, contrast responses in Tm9 ARE NOT LINEAR as they have pronounced activation for OFF stimuli but hardly any responses to ON stimuli. Given that this is the main point of the study, it is absolutely essential to show responses to individual bars. The authors never show a full response vs. contrast curve. To claim that Tm9 (and Tm2) are linear, they must show example responses at different contrast levels and provide a statistical measurement of linearity, which is not easy. How do you measure linearity with dF/F ? Is a stimulus that increases the calcium signal by 20% equal to a stimulus that decreases the calcium signal by 20%? What about a signal that increases the dF/F by 200%? Does it have an opposing intensity signal? Going back to point 1, is linearity even important?
>> As suggested by the reviewer, we are now showing actual calcium traces. In a new supplementary Figure (now Supplementary Fig. 2) we randomly selected 10 example Tm9 neurons and show that all Tm9 neurons show increases in calcium response to OFF bars and decreases in calcium response to ON bars (related to Fig. 2). We agree that claiming true linearity based on calcium responses is not trivial, but also not important for the claims of our paper. What is important is that Tm9 shows both, positive responses to OFF and negative responses to ON, and is described as such in the paper. We only use "linear" Tm9 responses for simplicity in the model in Fig. 1.

7. Figure 2: I found the color scheme to be unnecessarily confusing. The same color palettes are used for different stimuli (c, f) and RF properties (d, e). Also, why don't the authors show RF amplitude for ternary noise (figure 2f)?

>> We changed the color scheme in panels c and f, where we now use blue colors for all Tm9 responses, reflecting ON and OFF stimuli by the shade of blue. We also changed the color scheme throughout the paper to have one consistent color per neuron type. The filter obtained from ternary noise does not purely reflect a response, but the response multiplied with the stimulus. For example, negative responses to a negative contrast would give a positive filter contribution. Generally, filters can only be obtained up to a multiplicative constant (normalized, arbitrary units). This is now explained in the methods. Thus, one cannot not directly compare these filters to response amplitude to other stimuli such as bars.

8. Figure 3: it would be nice to see actual examples of responses to bars.

>> We added actual calcium traces imaged simultaneously in Tm4 and Tm9 (related to Figure 3) from ROIs from one example fly, these data are now shown in a new Supplementary Fig. 3.

9. Figure 4:

9.1. Why don't we get to see the data for Tm1 and Tm4 ON RF properties? It would add to the difference between ON responses across cells.

>> Tm1 and Tm4 ON RF properties were not shown, because no cells could be well fit by a Gaussian for quantification of parameters. As described in detail above (Reviewer #1, minor comment #29), we now show all data prior to selection in a new Supplementary Fig. 4. In brief, negative responses to ON are still prominent in Tm2 and Tm9 neurons, but not present in Tm4. Tm1 shows a negative peak in the tuning curve, mostly visible for vertical bars. However, because we shift traces (purely for visualization) after fitting a Gaussian for all ROIs, we cannot exclude that this is an artifact of alignment to the peak.

9.2. Why is the number of ROIs different between ON and OFF RFs? Figure label has a somewhat cryptic indication that 'Only neurons with responses in both orientations fitted by a single Gaussian with $r^2 > 0.2$ were included'. What does it mean? Did the authors exclude all ROIs that did not produce a good RF with ON bars? If that is the case, the data is extremely misleading! To give a hypothetical example: let's assume that a cell type always produces a good response (1 dF/F) to dark bars, but only 1% of neurons in that class respond to the ON bars (i.e., 99% of the cells have 0% change in dF/F and 1% reduce their responses by 1 dF/F). The natural conclusion from this example is that these cells prefer OFF signals, as their responses to OFF are 100 times larger than their responses to ON stimulation. However, if we don't count all the cells that had 0% change in dF/F (because the cell did not pass the 'Gaussian with $r^2 > 0.2$ ' inclusion criteria, we will reach the absurd conclusion that ON and OFF responses are represented equally by this population.

>> We apologize that we failed to mention this in the methods. The reason why much fewer ROIs were shown in the main figure is because we can only quantify parameters from a meaningful fit to the data, which we defined as a threshold of $r^2 \geq 0.2$ from fitting a Gaussian to each tuning curve for each orientation. Furthermore, a response quality index was used as a measure of trial-to-trial variability to exclude noisy cells regardless of the fit (after Baden et al. 2016 Nature). This is now both explained in detail in the methods.

ON responses are visible across all individual ROIs, including noisy ones. Even when averaging all responses ($n=98$ for Tm1, $n=152$ for Tm2, $n=55$ for Tm4, $n=108$ for Tm9), the mean amplitude of the negative ON response is similar to the ON response shown in the main figure (Supplementary Fig. 4).

10. Figure 5: I don't understand what is the purpose of this data. We know that Tm9 innervate T5 cells. Of course, they can influence T5 responses. Even the motivation (lines 191-195) makes it abundantly clear that Tm9 and T5 are linked. Would the manuscript suffer without this dataset? The only possible merit I see in such experiments is in the comparison between the effects of different presynaptic cells on T5 responses – but these experiments were not done.

>> Whereas Tm9 is the major T5 input with ~60 synapses, T5 also gets a total of 160 synapses from other neurons. We therefore still think that it is valid to investigate how a single presynaptic neuron might influence the responses downstream. Both Reviewer #2 ("a clever feature to demonstrate information flow from Tm9 to T5.") and Reviewer #3 commented positively on this. So we decided to leave this figure in the manuscript, and upon request of other reviewers moved further parts of the analysis from the Supplementary Figure into the main figure. To our knowledge, this is also the first time that variability within the responses of one cell type is described, and could potentially be interesting for the field.

11. Figure 6:

11.1. Panels a,b, it would help to add the zero line for dF/F traces.

>> We added the zero line (now Fig. 7a,b).

11.2. How can you get to $dF/F < -1$ (panel c). Where is the corresponding data in b?

>> The data in previous Fig. 6c (now Fig. 7c) originated from fitting difference-of-Gaussians to tuning curves in Fig. 6b (now Fig. 7b), consisting of a positive and a negative Gaussian located at the same position. The negative responses to ON can be best fitted by a larger negative Gaussian (center) and a smaller positive one (surround). Thus, an example tuning curve peaking at -0.5 could equally be fitted by two large values -3 and +2.5, or two smaller values -0.6 and +0.1. We originally did not strictly constrain the fitting procedure to avoid the larger values noticed by the reviewer. Based on this comment, we have imposed a constraint on the range of possible amplitudes each Gaussian can take, and re-generated the figure with the new fits. The main conclusions remain unchanged, namely CDM application sharpens Tm9 receptive fields by decreasing the amplitude of the OFF receptive field center and sharpening both ON and OFF receptive fields. We updated the figure description in the text accordingly. We additionally included a new Supplementary Fig. 6, where we quantify the maximum and the minimum responses, to observe the effects of CDM on response amplitudes independently from a parametric fit. The same supplementary figure also shows the quantification of receptive field sizes using a single Gaussian to fit the tuning curves, as done in previous figures. This provides support to the conclusions from the main figure.

11.3. Can the reduction of RF size after CDM be an artifact of smaller RF amplitude (the iceberg effect)?

>> We thank the reviewer for the remark and addressed the “iceberg effect” in a figure for the reviewer (Fig. R4). We normalized the tuning curves for OFF bars by their peak value, and see that the reduction of RF size remains visible even after discarding peak amplitude changes. Thus, the sharpening of the Tm9 OFF RF can be explained by a combination of a weaker center amplitude, and a sharper center.

Figure R4. Reduction of RF size upon CDM application after normalizing by peak amplitude. (Top) Receptive fields shown in Fig. 7b, normalized by their maximum value. Averages across ROIs and single ROI traces are shown. (Bottom) Mean normalized receptive fields overlapped to visualize receptive field size differences. Control (blue, n=46), CDM (orange, n=49).

11.4. Why are some cells responding with a negative dF/F to dark stimuli in f? Are they Tm9 cells? Were not the authors surprised to see this outcome? Why isn't it discussed?

>> As also described in a reply to reviewer 2: We apologize that we failed to point this, but we previously described this inverted response type in our initial characterization of Tm9 in Fisher et al. (2015) *Curr Biol* 25: 3178-89 (Figure S3). We now mention this in the figure legend: “*Few cells show an increase in calcium signal in response to OFF. This inverted response type likely represents cells not directly looking at the screen (see also Fisher et al. 2015).*”

12. Line 470: I am confused by this sentence. "response median" I assume it means 'the median value between different trials, is that correct? When was the median used, and when the maximum absolute response? Would not taking the max absolute response introduce noise or bias the measurement? For

example, if the responses' distribution is normal with mean=0, taking the absolute maximum would very likely result in a positive number for the cell response and not zero.

>> We corrected the methods to describe the procedure accurately. We always take the mean across trials. The “response median” was meant to address the median of the time course within a trial. However, the analysis now consistently uses the maximum absolute response to extract the tuning curve. This is explained in the methods: “To quantify the receptive field position, width and amplitude, we obtained a tuning curve as follows. The 1 s response traces to (shuffled) bar positions were averaged across trials and sorted by spatial coordinates. Taking the maximum absolute value of the response to each 1 s bar presentation resulted in a tuning curve. This tuning curve ...”

13. Almost everything in the manuscript is focused on the properties of the Tm9 cells. We get some glimpses on other excitatory cells in the system, but the contribution of their responses to T5 function is not measured (see point 10). Most importantly, the role of the main inhibitory cell CT1 in the computation of DS is not examined. The authors allude that CT1 is driven primarily by Tm9 and the reader is allowed to assume that CT1 responses are similar to Tm9. In fact, CT1 are cells with complex, long, narrow dendrites that are likely to innervate and receive inputs from numerous other neurons (Shinomiya et al 2019). How can the authors frame their study as 'we... identified the ON inputs that provide the cellular basis to implement a linear mechanism for direction selectivity in the fly OFF motion pathway.' (lines 244-5) when they clearly did not show any information about inhibition? Does CT1 provide an ON inhibitory input? Or maybe, it is a pure OFF cell? The latter is a definite possibility given that T5 is also postsynaptic to Tm9 - yet it is an OFF rectifying neuron. We know that inhibition is especially important in T4 and T5 DS computation, and thus this information is needed to support the conclusions of the study.

>> The reviewer correctly points out the importance of inhibition for T5 computation supported in literature (Gruntmann 2019). We have now included receptive field properties of CT1 neurons (Fig. 5) and show they are very narrow (~ 5 deg) consistent with findings in (Meier 2019). Furthermore, CT1 activity is reduced upon ON stimulation at the locations of the OFF receptive field center, and it is increased upon ON stimulation in the surround. Thus, T5 could respond negatively to OFF in its receptive field surround via CT1 inhibition, and respond positively to ON stimuli via CT1 release of inhibition. Furthermore the ON surround responses of CT1 could enhance the negative ON responses in the receptive field center of T5 by providing ON inhibition, in addition to the release of excitation from Tm2 and Tm9.

We did not directly measure if CT1 provides an inhibitory input to T5, but CT1 was shown to be Gad1-positive in Takemura et al. 2017. We also found CT1 repeatedly in genetic intersection of Gal4 lines from Silies et al. 2013 with a Gad1-reporter line (unpublished).

Reviewers' Comments:

Reviewer #1:

Remarks to the Author:

The authors have done a commendable job addressing my comments on the previous manuscript, and I think the revised paper is much improved. There are a few remaining issues that should be addressed before publication.

1) The title seems too broad, given the results in the paper. It implies that there is only one physiological basis ("*the* physiological basis") for direction-selectivity and that it's presented here. The basis explored here is one of many (for instance, the delays measured in Arenz et al., the almost-linear integration of inputs measured in Gruntman, etc.). This paper shows that the contrast-opponency they measure in theory enhances DS signals, but that does not make it the basis of direction-selectivity here. While the response properties upstream of T5 must be explained to figure out why T5 responds precisely as it does to contrasts, these physiological properties do not seem strictly required for DS responses (indeed, prior models of DS left it out). I think it would be better to make the title more specific to the result shown in the paper, for example highlighting non-preferred contrasts or contrast-opponency.

2) The authors say that the p-value is 0 if not a single one of a million shuffles generated an effect larger than the measured one. But it's not 0, it's just $<1e-6$, and that would be a correct way to report it.

3) Lines 565-6: LP and HP formulae. As written, the HP equation responds positively to *negative* derivatives, while the LP formula responds positively to *positive* deviations from 0. I believe there may be a sign error for the HP filter to make it the derivative of the LP filter. If these versions of the equations also existed in the code, then the high-pass inputs to the model would be oppositely signed to the LP inputs and the model would not be as stated (or intended). This should also be checked – I was unable to find any of the code at the Github link. An origin or citation for these functional forms would be helpful, since they don't look particularly standard. Can the authors also supply what the time constant was?

Minor

(Apologies for apparently missing some of these the first round.)

1) Line 11: "contrast-opponent RFs with ON and OFF subunits" – not clear what a subunit of a RF is. This terminology is primarily used when talking about nonlinear subunits upstream of a neuron that generate a particular RF, but here it's just being used to describe the shape (and sign) of a RF. Would "region" be more appropriate than "subunit"?

2) Lines 48 and 62: "linear summation" in T5 is still not totally clear what's being linearly summed: the inputs (spatially), in which case the inputs could still be rectified, or that the T5 response acts like a linear transformation of input contrasts, which must be accounted for in any proposed rectification. I think this could still be made clearer.

3) Line 142: "Tm9 provides ON information to T5" seems right. What does it mean for this to be required for T5? Maybe the authors mean "Tm9 provides required ON information to T5", since it's the information that's known to be required to explain T5 responses.

4) Line 183: "stronger" to "more strongly"

5) Fig 2a – scale bar for image?

6) Line 236: says recorded signals in LP, but should be lobula, no?

7) 4f and 4h – putting the horizontal line seems slightly misleading. How about text saying "none found"?

8) Line 272: "Ideally, ..." It's not clear why this ideal behavior is proposed. It's later referred to as

a hypothesis: can it just be framed as a hypothesis instead of proposing that this phenomenology is expected or the best?

9) Line 549 – This sentence claims that linear RFs are uncertain up to a scaling factor, citing Chichilnisky. This statement is only true when fitting LN models, which this paper does not do. I don't think the authors need to justify normalizing their RF amplitudes, and certainly should not use one that doesn't apply to their computation of linear filters.

10) Figure S4e certainly makes it look like Tm1 has negative responses to ON stimuli, but also has noisier measurements of individual ROIs. Is this a problem if the analysis isn't sensitive to this sort of average measurement (which seems apparent by eye) when individual traces are too noisy?

Reviewer #2:

Remarks to the Author:

All my concerns have been successfully addressed by the authors. I therefore recommend this work for publication.

Reviewer #3:

Remarks to the Author:

I would like to thank the authors for incorporating the reviewers' feedback. I consider the current version to be an adequate response to reviewers' concerns.

I do have a number of minor points/suggestions:

line 121: Fig. 1e,g, should be 1 f,g

I found the sentence on line 159 to be repetitive

line 183 I suggest adding 'respectively' at the end of the sentence.

fig 6h:p=0.0e0 should be 0

fig 7. label for f: I propose to replace 'not directly looking at the screen' with 'whose RF positioned outside of the boundaries of the screen' label for g: 'p-values were computed from permutation tests'

Point-by-point reply

We thank the reviewers for the overall positive assessment of our revised manuscript and addressed the remaining points below.

Reviewer #1 (Remarks to the Author):

The authors have done a commendable job addressing my comments on the previous manuscript, and I think the revised paper is much improved. There are a few remaining issues that should be addressed before publication.

1) The title seems too broad, given the results in the paper. It implies that there is only one physiological basis (“*the* physiological basis”) for direction-selectivity and that it’s presented here. The basis explored here is one of many (for instance, the delays measured in Arenz et al., the almost-linear integration of inputs measured in Gruntman, etc.). This paper shows that the contrast-opponency they measure in theory enhances DS signals, but that does not make it the basis of direction-selectivity here. While the response properties upstream of T5 must be explained to figure out why T5 responds precisely as it does to contrasts, these physiological properties do not seem strictly required for DS responses (indeed, prior models of DS left it out). I think it would be better to make the title more specific to the result shown in the paper, for example highlighting non-preferred contrasts or contrast-opponency.

>> We changed to title to “The physiological basis for contrast opponency in motion computation in *Drosophila* “

2) The authors say that the p-value is 0 if not a single one of a million shuffles generated an effect larger than the measured one. But it’s not 0, it’s just $<1e-6$, and that would be a correct way to report it.

>> Corrected as suggested .

3) Lines 565-6: LP and HP formulae. As written, the HP equation responds positively to *negative* derivatives, while the LP formula responds positively to *positive* deviations from 0. I believe there may be a sign error for the HP filter to make it the derivative of the LP filter. If these versions of the equations also existed in the code, then the high-pass inputs to the model would be oppositely signed to the LP inputs and the model would not be as stated (or intended). This should also be checked – I was unable to find any of the code at the Github link. An origin or citation for these functional forms would be helpful, since they don’t look particularly standard. Can the authors also supply what the time constant was?

>> We thank the reviewer for the observation, it indeed is a sign mistake only of the equation in the manuscript, not reflected in the code. During the revision of the manuscript we used the formulas given by Zavathone-Veth et al. 2020 for the LP/HP filters, and now added the citation to the methods. The code is now uploaded on Github.

Minor

(Apologies for apparently missing some of these the first round.)

1) Line 11: “contrast-opponent RFs with ON and OFF subunits” – not clear what a subunit of a RF is. This terminology is primarily used when talking about nonlinear subunits upstream of a neuron that generate a particular RF, but here it’s just being used to describe the shape (and sign) of a RF. Would “region” be more appropriate than “subunit”?

>> In response to the reviewer’s comment, we changed the word to “subfields”, as it is used in the reference of Leong et al. 2016.

2) Lines 48 and 62: “linear summation” in T5 is still not totally clear what’s being linearly summed: the inputs (spatially), in which case the inputs could still be rectified, or that the T5 response acts like a linear transformation of input contrasts, which must be accounted for in any proposed rectification. I think this could still be made clearer.

>> For more clarity, the sentence on line 48 now reads: “However, the cellular substrates that support the linear summation of presynaptic inputs onto T5, across space and time, have not been described.”

3) Line 142: “Tm9 provides ON information to T5” seems right. What does it mean for this to be required for T5? Maybe the authors mean “Tm9 provides required ON information to T5”, since it’s the information that’s known to be required to explain T5 responses.

>> Line now reads “Tm9 provides ON information to T5”

4) Line 183: “stronger” to “more strongly”

>> Done.

5) Fig 2a – scale bar for image?

>> Added.

6) Line 236: says recorded signals in LP, but should be lobula, no?

>> We thank the reviewer for catching this mistake, and corrected the text.

7) 4f and 4h – putting the horizontal line seems slightly misleading. How about text saying “none found”?

>> Done.

8) Line 272: “Ideally, ...” It’s not clear why this ideal behavior is proposed. It’s later referred to as a hypothesis: can it just be framed as a hypothesis instead of proposing that this phenomenology is expected or the best?

>> Following the suggestion, we replaced “Ideally, ...” with “We hypothesize that ...”.

9) Line 549 – This sentence claims that linear RFs are uncertain up to a scaling factor, citing Chichilnisky. This statement is only true when fitting LN models, which this paper does not do. I don’t think the authors need to justify normalizing their RF amplitudes, and certainly should not use one that doesn’t apply to their computation of linear filters.

>> We removed the justification and corresponding reference.

10) Figure S4e certainly makes it look like Tm1 has negative responses to ON stimuli, but also has noisier measurements of individual ROIs. Is this a problem if the analysis isn’t sensitive to this sort of average measurement (which seems apparent by eye) when individual traces are too noisy?

>> The responses of individual ROIs are indeed noisier, have larger trial-to-trial variability, and consequently do not pass the threshold for the response quality index. The analysis is done on an ROI basis, and we consider it misleading to quantify the unreliable single cell responses, with high trial variability and low goodness of fit. While the average trace across ROIs shows a negative response, this trace is produced solely for visualization by aligning the traces to the receptive field center position obtained from the (low quality) fits, which could itself bias the mean to have some peak that is not actually reflecting the data. We prefer to error on the more conservative side, and would not like to make claims unsupported by data.

Reviewer #2 (Remarks to the Author):

All my concerns have been successfully addressed by the authors. I therefore recommend this work for publication.

Reviewer #3 (Remarks to the Author):

I would like to thank the authors for incorporating the reviewers' feedback. I consider the current version to be an adequate response to reviewers' concerns.

I do have a number of minor points/suggestions:

line 121: Fig. 1e,g, should be 1 f,g

>> Corrected.

I found the sentence on line 159 to be repetitive

>> The sentence "Quantification of the spatial receptive field showed that Tm9 ON responses were variable, ranging from 8.3 deg to 71.5 deg (mean 28.1 deg, std 17.6 deg, grouping both orientations)." now reads "Quantification of the spatial receptive field showed that Tm9 ON responses were variable, ranging from 8.3 deg to 71.5 deg."

line 183 I suggest adding 'respectively' at the end of the sentence.

>> Done.

fig 6h:p=0.0e0 should be 0

>> We have corrected this to $1 < 1e-6$ following the observation from reviewer #1.

fig 7. label for f: I propose to replace 'not directly looking at the screen' with 'whose RF positioned outside of the boundaries of the screen' label for g: 'p-values were computed from permutation tests'

>> Label 7f "Few cells show a decrease in calcium signal in response to OFF. This inverted response type likely represents cells not directly looking at the screen (see also 14)." now reads "Few cells show a decrease in calcium signal in response to OFF, likely because their receptive fields are positioned outside the boundaries of the screen (see also 14)."
Label 7g now reads "P-values were computed from two-tailed permutation tests".